# Noncontact human-machine interaction based on hand-responsive infrared structural color

Shun An [1,3], Hanrui Zhu[1,3], Chunzhi Guo[2], Benwei Fu[1], Chengyi Song [1], Peng Tao [1], Wen Shang [1✉] &
Tao Deng [1✉]

Noncontact human-machine interaction provides a hygienic and intelligent approach for the communication between human and robots. Current noncontact human-machine interactions are generally limited by the interaction distance or conditions, such as in the dark. Here we explore the utilization of hand as an infrared light source for noncontact human-machine interaction. Metallic gratings are used as the human-machine interface to respond to infrared radiation from hand and the generated signals are visualized as different infrared structural colors. We demonstrate the applications of the infrared structural color-based human-machine interaction for user-interactive touchless display and real-time control of a robot vehicle. The interaction is flexible to the hand-interface distance ranging from a few centimeters to tens of centimeters and can be used in low lighting condition or in the dark. The findings in this work provide an alternative and complementary approach to traditional noncontact human-machine interactions, which may further broaden the potential applications of human-machine interaction.

[1] State Key Laboratory of Metal Matrix Composites, School of Materials Science and Engineering, Shanghai Jiao Tong University, 800 Dongchuan Road, Shanghai 200240, China. [2] School of Electronic Information and Electrical Engineering, Shanghai Jiao Tong University, 800 Dongchuan Road, Shanghai 200240, China. [3] These authors contributed equally: Shun An, Hanrui Zhu. ✉email: shangwen@sjtu.edu.cn; dengtao@sjtu.edu.cn

With the rapid development of artificial intelligence, human–machine interaction (HMI) plays an important role in the seamless communication between human and robots[1]. HMI has shown promise in various applications, including healthcare and medical assistance[2–4], smart home and vehicle[5–7], and virtual reality[8–11]. In HMIs, the bridge between human and machines is the human–machine interface that can sense and convert human stimuli into executable signals, which are then transmitted to machines[12]. The human–machine interfaces in general are pressure-sensing[13–16], strain-sensing[17–20], or surface electromyography sensing[21–23]. For these HMIs, the human body needs to be in touch with the interface to enable sensing and transmission, which is normally inconvenient and leads to inevitable mechanical wear and fatigue[2]. The frequent contact of the interface also increases the risk of cross-infection of bacteria and viruses between users, especially for the applications in healthcare and medical assistance[24]. Noncontact HMI thus shows unique advantages over contact HMI in the aspects of convenience, long operating life, and hygiene.

The humidity around the finger, the charges gathered at the fingertip, and the thermal energy of the human body have been reported to be used as stimuli for noncontact HMI[2,25–27]. These stimuli, however, are effective only within a limited distance, which is generally less than 2–3 cm. For example, Park et al. developed an HMI platform that is responsive to the humidity around a finger[25]. The platform was demonstrated to be effective for intelligent and touchless display. The need for humidity, however, makes such an HMI interface less effective when the fingers are more than 2 cm away from the platform and also slows the response time of the interaction to ~20 s. Direct vision-based gesture recognition is another effective approach for noncontact HMI, but the efficiency of this method is limited by the self-occlusion of complex gestures and environment interference such as the poor light condition[28,29]. The exploration of alternative human stimuli to enhance the flexibility and robustness of HMI may further enable the intelligent interaction between human and machines.

With the body temperature of human stabilized at around 37 °C, the human hand naturally emits infrared (IR) radiation and therefore can be used as a powerless and intelligent IR light source. With each finger acting as an independent IR light source, complex signals can be generated by different gestures formed by fingers[30]. Specifically, the IR radiation from the hand is mainly within the atmospheric transmission window (7.5–14 μm), which means the IR radiation can transmit in the air and then interact with physical structures[31,32]. In this work, we explored the response of metallic gratings to IR radiation from human hand for noncontact HMI. We first explored the effect of grating periods and duty cycles on the IR diffraction intensity with the whole hand as an IR light source. The diffracted IR radiation with different intensities can be visualized as different colors shown by the IR detector, which characterize the specific interaction between hands and different gratings. We define the generated colors as IR structural colors and such IR structural colors were used as the effective signals for HMI in this work. The conversion of IR diffraction intensities into IR structural colors realizes the direct visualization of the interaction between hand and different gratings, which is friendly for users to determine the relative position between hand and grating and judge whether the generated signal is correct. We can place the hand close to the grating interface for user-interactive touchless display and multilevel information encryption/decryption. Besides the whole hand, the finger can also act as an IR light source for HMI. As shown in Fig. 1, the finger at a specific position can selectively interact with one specific grating and generate desired IR structural color. The

structural color can be recognized as a specific command for HMI. With the movement of the finger, different IR structural colors can be generated, which can be recognized as different HMI commands. As a proof-of-concept demonstration, we used only one finger to command a robot vehicle to perform complex actions in real-time and also in the dark. The findings in this work provide an alternative approach for noncontact HMI with flexible hand-interface distance and high robustness even in the dark condition, which may further broaden the potential applications of HMI.

## Results

**Generation of static IR structural colors using the human hand.** Metallic reflective gratings could diffract IR radiation strongly. When one beam of incident light with a specific wavelength interacts with the gratings, the incident light is diffracted into discrete directions with specific diffraction angles (Fig. 2a). The relationship between the incident angles and diffraction angles satisfies the diffraction equation[33]:

$$\sin \theta_{\mathrm{in}} = \sin \theta_d - \frac{m\lambda}{\Lambda} \qquad (1)$$

where $\theta_{in}$ is the incident angle, $\theta_d$ is the diffraction angle, $m$ is the diffraction order and is a nonzero integer, $\lambda$ is the wavelength of the light, and $\Lambda$ is the grating period. Equation 1 shows that for the incident light with a specific wavelength, the relationship between incident angle and diffraction angle is mainly determined by the grating period. Besides the angles, the diffraction efficiency ($\xi$, the ratio of IR diffraction intensity to the incident IR intensity) is another important parameter, which depends on not only the grating periods but also the duty cycles ($\eta$, the ratio of line width to period)[34].

Human hand is a natural IR emitter with a radiation peak at ~9 μm based on the Planck distribution function[30,35]. To study the interaction between the IR light from hand and metallic gratings, we fabricated aluminum (Al) grating arrays with periods from 25 to 75 μm and duty cycles from 10 to 90% by photolithography. Figure 2b shows some of the gratings with different periods and duty cycles. The thickness of these gratings is ~3 μm (Supplementary Fig. 1). An IR detector (FLIR T620) was used to capture the diffracted IR radiation from different gratings. The IR detector was set at ~10° from the axis perpendicular to the center of the sample (Supplementary Fig. 2), which means the detected diffraction angle ($\theta_d$) was 10°. For the fixed diffraction angle and specific period, the corresponding incident angle can be calculated by Eq. 1, which is related to the light wavelength ($\lambda$). Since the detected wavelength range by the IR detector is 7.5–14 μm, the corresponding incident angle is a range rather than a specific value. Each angle corresponds to one specific light wavelength.

We placed the hand at the side of the sample (Supplementary Fig. 2) with the incident angle range of 11°–90°, which covers the incident angle ranges for all the fabricated gratings with different periods. The IR radiation from the hand is diffracted by all the gratings simultaneously while with different efficiencies, which leads to different IR diffraction intensities. The change of IR diffraction intensities can be shown as color distribution in the IR image (Supplementary Note 1). Despite the color in the IR image does not represent the intrinsic "color" or wavelength of IR radiation, it can characterize the light intensity after the specific interaction between IR radiation and different gratings. The color shown on the IR detector can thus act as IR structural color for different grating structures. As shown in Fig. 2c, the change of IR diffraction intensities from different gratings showed rainbow colors (red, yellow, green, cyan, blue, and violet), which yielded an IR color palette. Here we used the normalized IR diffraction intensity to show the differences of IR diffraction intensities from

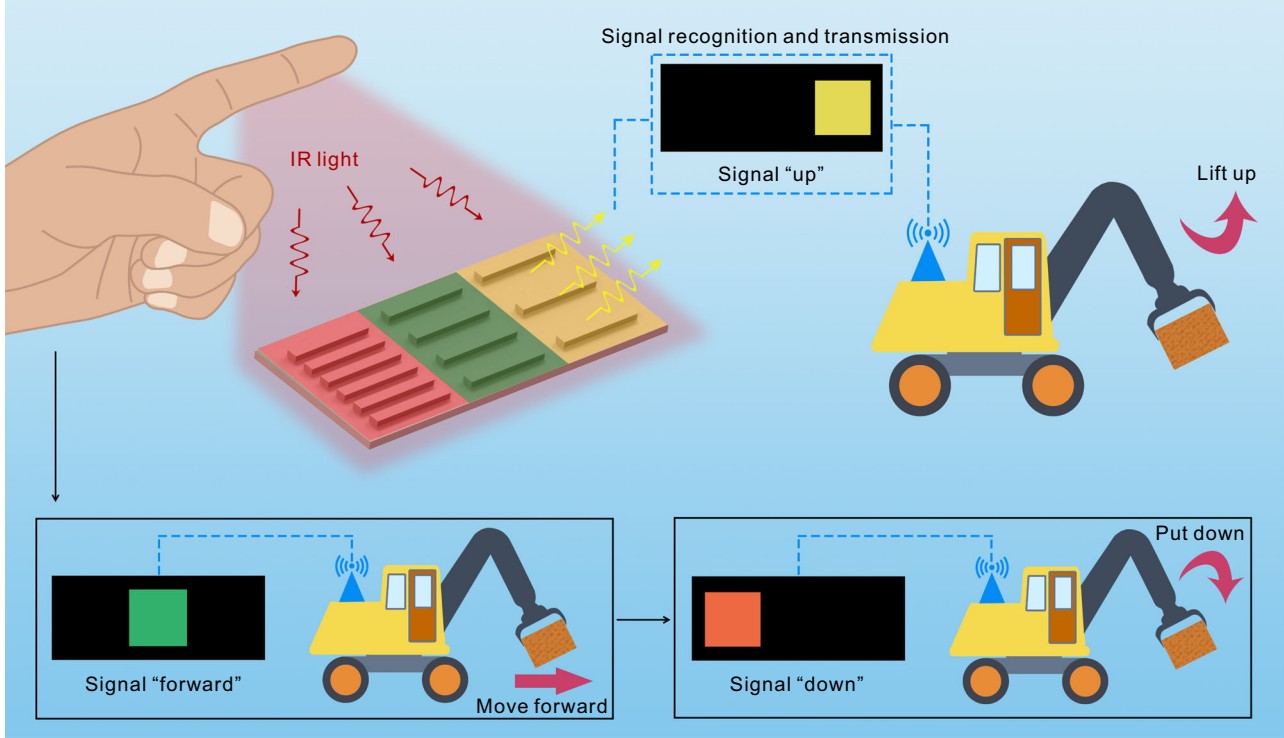

**Fig. 1 Schematic illustration of the noncontact HMI based on hand-responsive IR structural color.** The IR radiation from a finger at a specific position can selectively interact with specific metallic grating by diffraction. The diffracted IR radiation from the specific grating can be detected as specific IR structural color and then recognized as the corresponding command to control the action of a robot vehicle. With the movement of the finger, different IR structural colors can be generated, which can be recognized as different commands to control the robot vehicle.

different gratings (Supplementary Note 1). When there was no hand, the normalized IR diffraction intensities from different gratings were all zero, which is shown as the color black (Supplementary Fig. 3). We can pick gratings with different IR structural colors from the IR color palette for user-interactive touchless display. As shown in Fig. 2d, we used six different gratings (dashed boxes in Fig. 2c) to generate an IR color pattern of a house. The SEM images of these gratings are shown in Supplementary Figure 4. When the human hand was placed at the side of the "house", the "house" image captured by the IR camera exhibited different colors for different components, with the violet chimney, blue roof, green wall, cyan window, yellow door frame, and red door. The generated IR structural colors can be used not only for color display but also for multilevel color anti-counterfeiting (Supplementary Fig. 5), which can add another level of security compared to monochromatic anti-counterfeiting.

From the IR color palette in Fig. 2c, we can find that the duty cycles play a more important role in the generation of different IR structural colors than the grating periods. We further quantitively studied the effect of duty cycles on the IR diffraction intensity. As shown in Fig. 2e–g, with the specific grating period (25, 50, or 75 μm), the IR diffraction intensity increases first with the increase of the duty cycle and then decreases. The maximum IR diffraction intensity is at the duty cycle of ~60%. We further used the finite-difference time-domain (FDTD) method to numerically calculate the IR diffraction intensity for different gratings (Supplementary Note 2). As shown in Fig. 2h–j, the calculated change of IR diffraction intensity with duty cycles is in good agreement with the experimentally measured change of IR diffraction intensity with duty cycles.

**Finger-responsive dynamic IR structural colors**. In the above demonstration, the hand was placed close to the grating interface at

a specific position for static color display. Since the IR radiation can transmit in the air freely, we can also move the hand for the generation of dynamic IR structural colors. To explore the dynamic interaction between hand and structural colors, we fabricated a word of "yes" by the Al gratings with the period of 40 μm and the duty cycle of 60% and used the index finger as the IR light source (Fig. 3a, b). With the detected diffraction angle of $\theta_d = 10°$, the corresponding incident angle range for −1st diffraction is 21°–32° according to Eq. 1. When the finger is close to the gratings, it can cover the whole incident angle range. With the finger moving away from the gratings, the covered incident angle range decreases gradually (Fig. 3a), which will lead to the decrease of the IR diffraction intensity. In this study, we first placed the finger at one position close to the gratings (position 1, P1 in Fig. 3a) and then moved the finger to P2 and then P3. With the finger moving away, the word "yes" exhibited different IR structural colors (Fig. 3c). In the interaction between the finger and the grating, the other parts of the hand and the background also emit IR radiation and interact with the grating. Due to the fix of the detected diffraction angle, not all the IR radiation (from hand and from the background) can be detected by the IR camera after the interaction with gratings. Only the IR radiation with incident angle satisfying diffraction equation (Eq. 1) can interact with gratings and then the diffracted light is detected by the IR camera. The selective interaction between finger and grating minimizes the interference of IR radiation from other parts of the hand and the complex background on the generation of IR structural colors. The intelligent generation of dynamic IR structural colors can be employed in HMI for the dynamic signal transfer through recognition of dynamic motions related to the finger. For example, when the color of the word changes from red to green by moving the finger away from the sample, it can represent a specific executable command. When the color of the word changes from green to red by moving the finger towards the sample, it can represent another executable command.

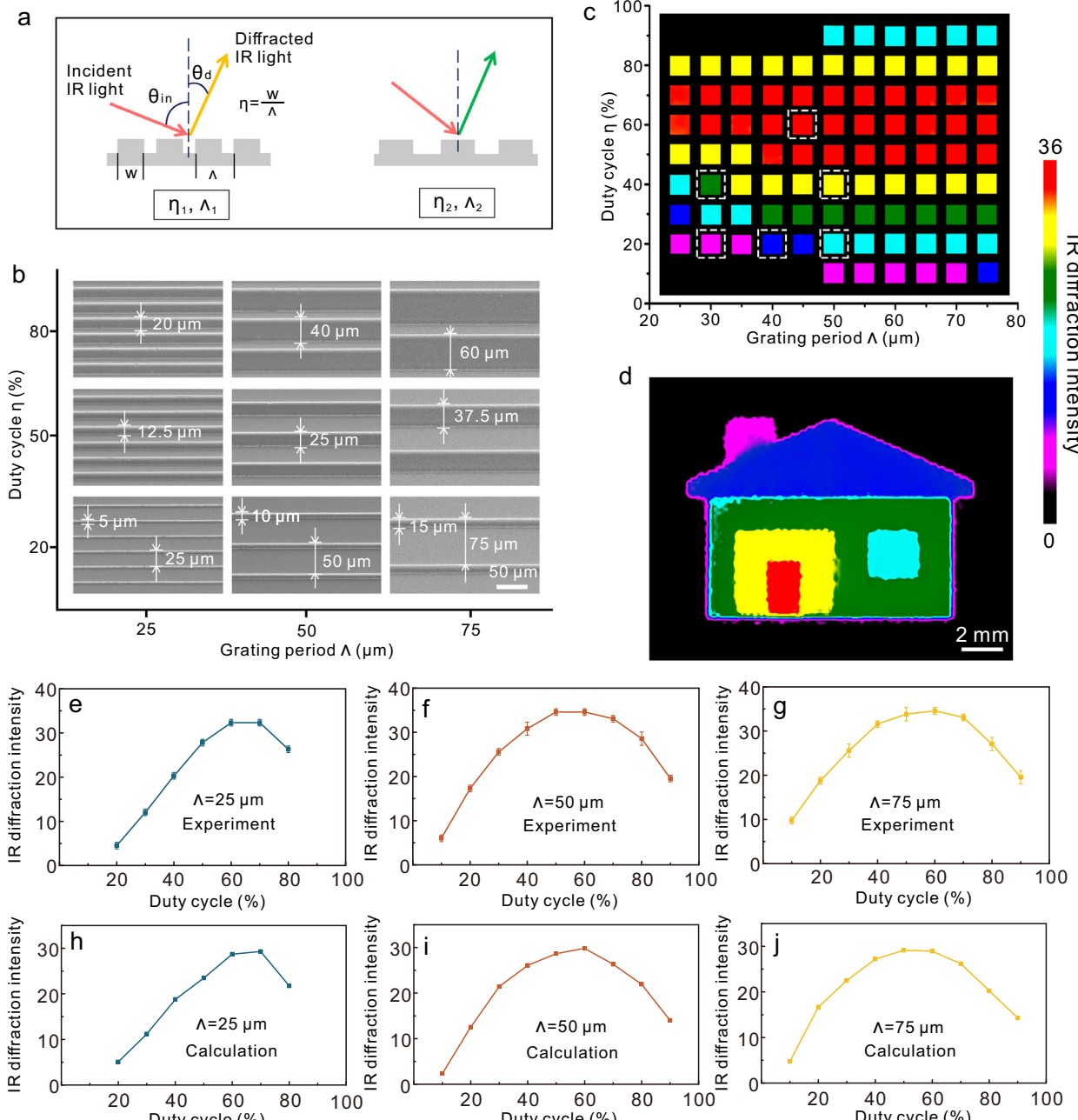

**Fig. 2 The response of gratings with different structural parameters to human hand. a** Schematic illustration of the interaction between light and gratings with different periods ($\Lambda$) and duty cycles ($\eta$). **b** SEM images of fabricated Al gratings. **c** The IR color palette generated based on different grating periods and duty cycles. **d** Color display of a "house" based on the selected gratings (dashed boxes) in Fig. 2c. **e–g** Experimentally measured IR diffraction intensity with different duty cycles. Error bars represent the standard deviation of the mean. **h–j** FDTD calculated IR diffraction intensity with different duty cycles.

With the flexibility of adjusting the position of the finger, the finger can selectively interact with multiple different gratings for the generation of dynamic structural colors. The diffraction equation (Eq. 1) shows that the incident angle ranges are different for gratings with different periods. With the smaller grating period, the incident angle for −1st diffraction angle $\theta_d = 10°$ ($m = −1$) is larger for one specific wavelength. As shown in Fig. 3d, compared to the grating (Gy) in the middle, the incident angle range (the red triangle) for the grating (Gr) at the left with a smaller period is closer to the substrate. The incident angle range (the green triangle) for the right grating (Gg) with a larger period on the other hand is further away from the substrate. When the finger is placed within range 1 (R1), which is included in the

incident angle range of Gr and excluded in the incident angle range of Gy and Gg, the IR detector will capture the IR structural color from Gr but not from Gy and Gg. If the finger moves to R2, which is included in the incident angle range of Gy and excluded in the incident angle range of Gr and Gg, the IR detector will capture the IR structural color from Gy but not from Gr and Gg. Similarly, when the finger further moves up to R3, which is included in the incident angle range of Gg and excluded in the incident angle range of Gr and Gy, the IR detector will capture the IR structural color from Gg but not from Gr and Gy. Besides the selective interaction between the finger and different gratings, the generated structural colors differ from each other as well due to the difference of duty cycles for different gratings.

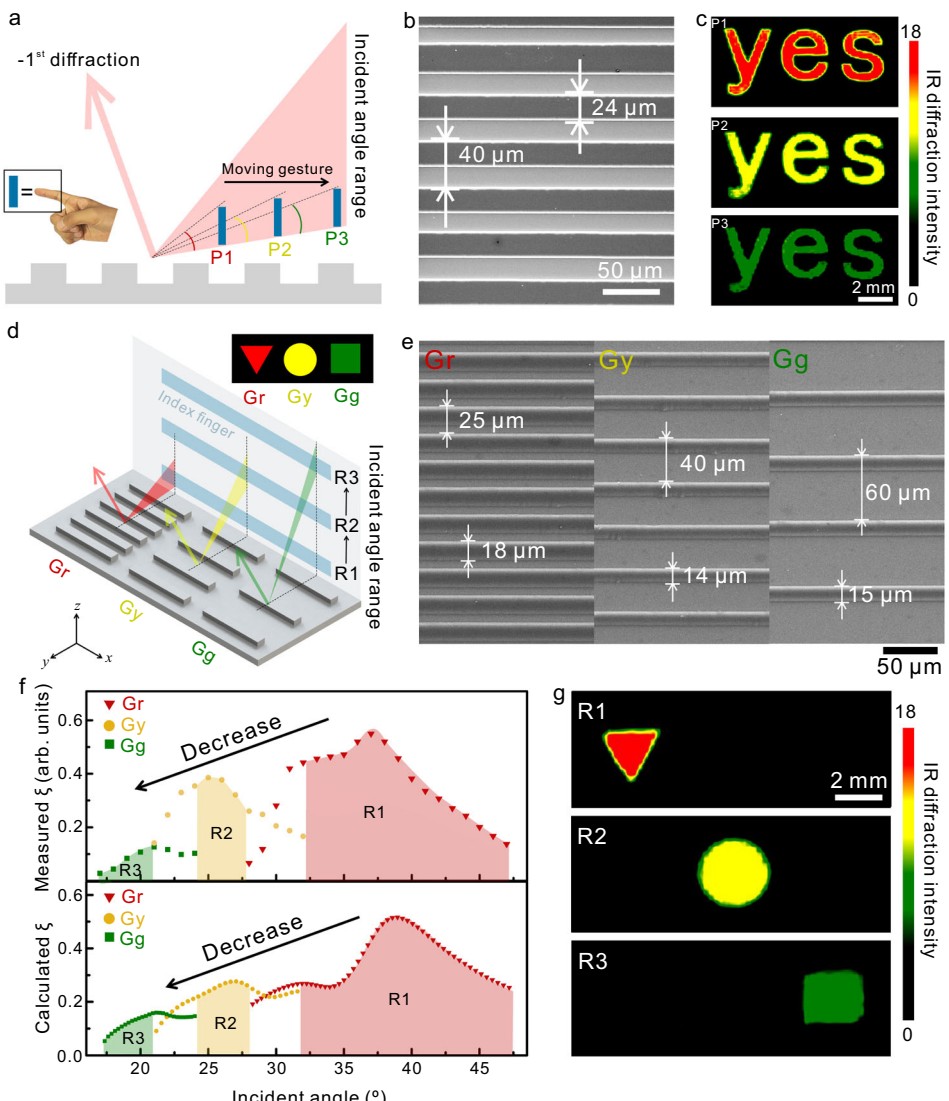

**Fig. 3 Interaction between the finger and gratings for the generation of dynamic IR structural colors. a** Schematic illustration of the interaction between the finger and one single grating. **b** The SEM image of the fabricated gratings. **c** IR images generated through moving the finger away from the gratings. **d** Schematic illustration of the interaction between the finger and different gratings; The red, yellow, and green areas represent the incident angle ranges of Gr, Gy, and Gg, respectively. **e** SEM images of gratings. **f** Experimentally measured −1st diffraction efficiency (arbitrary units, arb. units) of different gratings (top) and FDTD calculated −1st diffraction efficiency of different gratings (bottom). **g** IR images generated by moving the finger vertically.

To demonstrate the selective interaction between the finger and different gratings for dynamic generation of IR structural colors, we used three different gratings (Gr with the period of 25 μm, Gy with the period of 40 μm, and Gg with the period of 60 μm) to generate a pattern of different shapes with different colors (Fig. 3d, inset and Fig. 3e). The corresponding incident angle ranges for the three gratings are 28°–47°, 21°–32°, and 17°–24°, respectively. We can find that the incident angle ranges are partially overlapped. To realize the selective interaction between fingers and different gratings, R1–R3 should be only within the incident angle range of one grating without overlapping with each other. R1–R3 are thus specified to 32°–47°, 24°–28°, 17°–21°, respectively. We first experimentally measured the distribution of diffraction efficiency for the three gratings (Supplementary Figure 6 and Supplementary Note 3) and then used FDTD to calculate the distribution of diffraction efficiency for the three gratings. As shown in Fig. 3f, both experimental measurement and theoretical calculation show that diffraction efficiency is continuously varied with the incident angle. R1 has the highest diffraction efficiency, followed by R2, and R3 has

the lowest diffraction efficiency. When we placed the finger in R1, the IR radiation from the finger selectively interacted with Gr and generated the structural color with the highest IR diffraction intensity. The Gr grating was thus shown in red. When we moved up the finger to R2 or R3, the IR radiation selectively interacted with Gy or Gg to exhibit the corresponding pattern in yellow or green, respectively (Fig. 3g). In the above demonstration, the index finger was used as the IR light source with the finger-interface distance of 10–15 cm. When the whole hand was used as the IR light source for the generation of dynamic IR structural colors (Supplementary Fig. 7), the hand can interact with the grating interface from a longer distance (30–50 cm) since the area of the whole hand is much larger than that of a finger. The freedom of changing both the position and gesture of the hand provides the flexibility for the adjusting of the interaction and also the possibility of the generation of more complicated IR structural color images for complex interactions.

Hand perspiration has little effect on the interaction between hand and grating arrays due to the low IR reflectivity of $H_2O$ film

on the hand and the high IR transmissivity of $H_2O$ vapor around the hand (Supplementary Fig. 8)[36]. The interaction between hand and grating arrays is applicable for different environmental temperatures (Supplementary Fig. 9a–e). Specifically, when the environmental temperature is higher than the hand temperature (Supplementary Fig. 9e), the IR radiation intensity of the hand is lower than that of the environmental background. In this case, the IR diffraction intensity of the grating is decreased when the hand is used to interact with the grating (Supplementary Note 1). The color bar is thus inversed to characterize the generated IR structural colors. When the environmental temperature is the same as the hand temperature, the IR radiation from the hand cannot cause the change of IR diffraction intensity of gratings and thus cannot generate different IR structural colors. In this case, we can rub the hand to increase the hand temperature instantly, and thus increase the temperature differences between the hand and the environment (Supplementary Fig. 9f). In the situations that need to involve other IR light sources rather than human hand for the HMI, in the Supplementary Information, we showed that other IR light sources, for example, a heating plate with the temperature of ~36 °C, a smartphone with the temperature of ~40 °C, and a bottle of warm water with the temperature of ~45 °C, can also be used as IR light sources for dynamic generation of IR structural colors for possible use in HMI (Supplementary Figs. 10–12). The grating array can also be fabricated by soft lithography, in which a flexible (polydimethyl-siloxane, PDMS) stamp is first generated and then used to transfer microstructures onto other substrates. The stamp can be used multiple times and the fabrication is cost-effective for large-scale applications[37]. In this work, we also demonstrated the use of soft lithography for the fabrication of grating arrays (Methods and Supplementary Fig. 13). As shown in Supplementary Fig. 14, gratings with different structural parameters on Si substrate were fabricated by soft lithography. The PDMS stamp can also be used directly as a flexible grating. We fabricated a PDMS grating with a period of 45 μm and a duty cycle of 50%. The incident angle range of the grating is 20°–29°. Placing the index finger in the incident angle range and moving the finger resulted in different IR structural colors (Supplementary Fig. 15, top). We stretched the sample along the direction of periods by 20%, which means the period of the grating would increase to 54 μm. and the corresponding incident angle range would change to 18°–26°. When the finger was placed in the incident angle range of 18°–26°, it can interact with the grating to generate IR structural colors as well (Supplementary Fig. 15, bottom).

**Demonstration of real-time HMI**. With the selective interaction between the finger and gratings, one finger can generate different IR structural colors, which can be used for real-time HMI. As shown in Fig. 4a, we used the three gratings in Fig. 3e to generate a pattern of the traffic light. When the index finger was placed in the incident angle range of Gr (P1), the Gr grating was shown as red light (Fig. 4a, b). With the index finger moving away to P1', the red light from Gr turned to yellow light. Further moving the index finger to P1", the yellow light from Gr turned to green light. The index finger can also easily move in the vertical direction. When the index finger moved from P1 to the incident angle range of Gy (P2), the red light of Gr grating turned off and the Gy grating was shown as yellow light. When the index finger further moved to the incident angle range of Gg (P3), the yellow light of Gy grating turned off and the Gg grating was shown as green light. We can thus use only one finger to generate five different patterns of IR structural colors, which are the combination between colors and the positions of color signals (Fig. 4b). The generated patterns of IR structural colors can be recognized as

different commands to control the actions of a robot vehicle (Supplementary Note 4). In the proof-of-concept demonstration, the task of the robot vehicle is to transport an object to the destination (Fig. 4c). The robot vehicle first needs to lift up the object and then move to the destination. At the destination, the vehicle puts down the object. As shown in Fig. 4d and Supplementary Movie 1, the robot vehicle can follow the commands based on the generated patterns of IR structural colors and complete the task. The IR intensity resolution of the device is represented by the temperature resolution of the IR detector used in the HMI process, which is ~0.1 °C. The response time is 20.96 ± 1.20 ms, which is the time from the IR detector starting to capture the pattern of IR structural colors to sending the recognized command to the robot vehicle.

Since the visible light has no interference on the generation of IR structural colors, the robot vehicle can also execute the task in the dark, which demonstrates the high robustness of this HMI system in low visible lighting conditions or even in complete darkness (Supplementary Movie 2). In a dark environment, we first move the finger to generate one pattern of IR structural color. Based on the generated IR structural color, we can determine the relative position between the finger and grating array and then move the finger to generate other patterns of IR structural colors. The grating array can also be integrated into a wrist band and attached to the wrist to fix the relative position between fingers and gratings and thus enhance the reliability and operability of HMI in the dark. As shown in Fig. 4e, two same gratings with a period of 20 μm and a duty cycle of 50% are attached on a wrist band and aligned with the middle finger and the ring finger, respectively. The two gratings are invisible in the visible image of the wrist band so we use two squares with different colors to show the relative position of the gratings (Fig. 4e). The grating corresponding to the blue square is named Gb and the grating corresponding to the violet square is named Gv. With the bending of a middle finger, the IR radiation from the middle finger will selectively interact with Gb. The covered incident angle range is adjustable by bending the finger with different degrees. Similarly, the ring finger can selectively interact with Gv by bending the finger. Different IR color patterns are generated with the bending of different fingers (Fig. 4f and Supplementary Movie 3). The recognition accuracy of the wrist band is 99.2% and the wrist band is applicable for operators with different hand lengths (Supplementary Fig. 16). In the future, IR detectors may also be integrated into the wrist band to fix the relative position among fingers, grating array, and IR detectors, which can further enhance the reliability and portability of the HMI system.

**Comparison between direct-gesture based HMI and IR structural color-based HMI**. IR camera is widely used to directly capture the IR images of gestures. The captured gestures are recognized as specific commands for HMI, which can be referred to as direct gesture-based HMI. Different from the direct capture and recognition of complex gestures by IR camera, in this work we use gratings to serve as the interface for the selective inter-action with the IR light from fingers and to convert complex gestures into easily recognizable patterns of IR structural colors. We further compare gesture-based HMI and IR structural color-based HMI from four aspects: (1) Amount of information captured for processing. The hand is a complex articulated object with more than 20 degrees of freedom[28]. The gestures are thus complex and IR images of gestures contain a large amount of information, which makes real-time HMI challenging. In comparison, the IR structural color patterns contain much less amount of information, which is beneficial for real-time recognition (Supplementary Fig. 17 and Supplementary Note 5). (2)

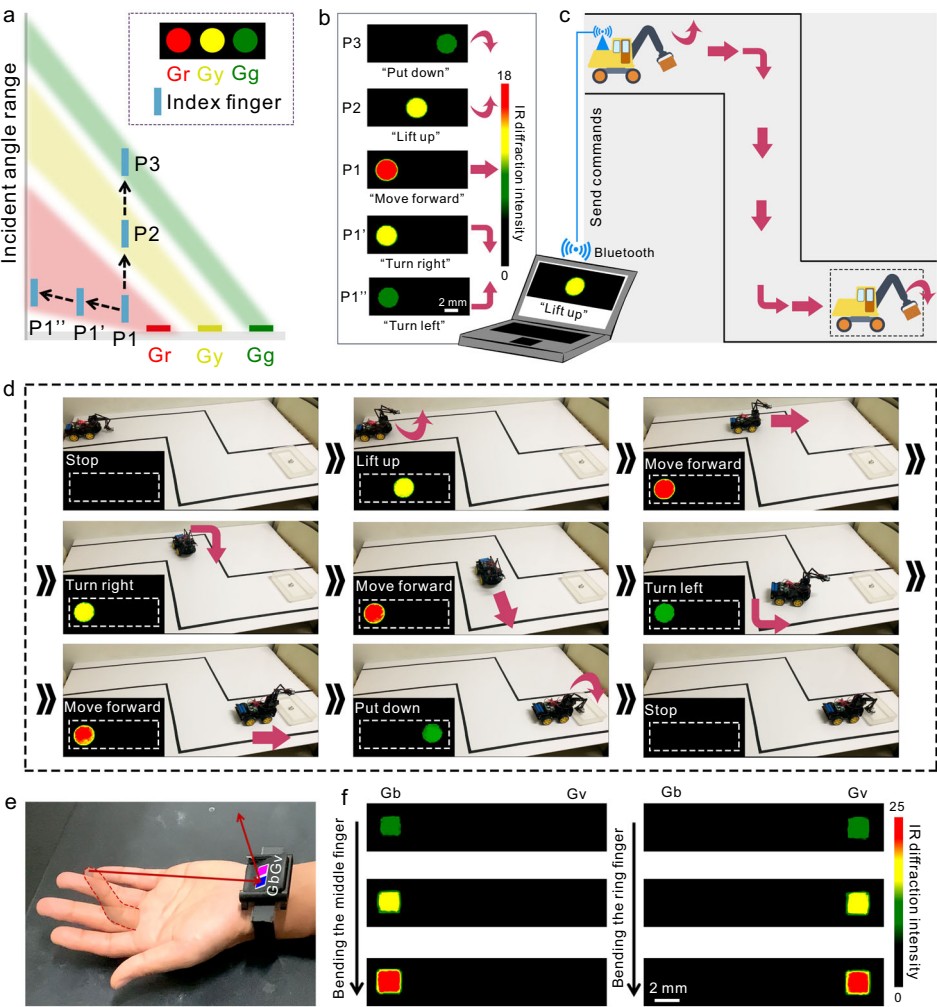

**Fig. 4 Demonstration of real-time HMI based on finger-responsive IR structural colors. a** Schematic of the positions of the finger for the generation of multiplexed IR structural colors. **b** Generated patterns of IR structural colors and the corresponding recognized commands. The commands are recognized based on the combination of both IR structural colors and the positions of colors. **c** Schematic illustration of the task of the robot vehicle in the proof-of-concept demonstration. The commands are sent to the Bluetooth on the robot vehicle to perform different actions. **d** Real-time actions of the robot vehicle based on the finger-responsive IR structural colors. **e** Two gratings (blue grating (Gb) and violet grating (Gv)) are integrated into a wrist band, which are aligned with the middle finger and the ring finger respectively. **f** The generated IR color patterns with the bending of the middle finger/ring finger.

**Interference of background.** In direct gesture-based HMI, the complex background will interfere with the segmentation of gestures and thus affect the recognition performance[19]. In comparison, this IR structural color-based HMI uses the grating array as the interface to selectively interact with fingers, which minimizes the interference of the background and other parts of the hand (Supplementary Fig. 18). (3) Interference of self-occlusion. In direct gesture-based HMI, different gestures are distinguished from each other by their unique features. The self-occlusion, which means the block or overlap of fingers by other fingers or other parts of the hand, is hard to avoid and interferes with the extraction of features[38]. In IR structural color-based HMI, the highly selective interaction between an individual finger and the corresponding grating minimizes the impact of self-occlusion, since grating only interacts with the corresponding finger at a specific position, while other fingers or other parts of the hand will not interfere with such interaction (Supplementary Fig. 19). (4) Algorithm. Due to the interference of complex background and self-occlusion, the gesture recognition process is highly dependent on algorithms and complex algorithms are needed for the recognition process. In comparison, IR structural color-based

HMI uses a grating array to interact with the hand. The feature of each gesture is directly converted into easily recognizable patterns of IR structural colors. The use of a grating array thus simplifies the algorithm to directly recognize features of images, with no need for gesture segmentation from the background and hand modeling and feature extraction. In summary, the selective interaction between hand and gratings in IR structural color-based HMI helps minimize the interference of background and the self-occlusion of the hand. The generated simple and recognizable patterns of IR structural colors also generate much fewer data to be processed and require less complex algorithms in the analysis process, both of which help simplify the signal recognition process. The detailed comparison result is shown in Supplementary Note 6.

## Discussion

In summary, this work demonstrates a noncontact HMI system with high flexibility and robustness based on the hand-responsive IR structural colors. Gratings with different periods and duty cycles can diffract IR radiation with different intensities, which

can be shown as the color distribution in the IR images. The color distribution characterizes the interaction between IR radiation and different grating structures, that is, IR structural colors. An IR color palette was generated in this work to characterize the IR structural color for different gratings. We can pick gratings with specific IR structural colors from the palette for user-interactive touchless display. The experimental result and numerical calculation demonstrate that for different gratings the IR diffraction intensity increases first with the increase of the duty cycle and then decreases. The maximum IR diffraction intensity is at the duty cycle of 60%. On the other hand, for the same grating structure, with the finger moving away from the gratings, the finger covers a smaller range of incident angles, which decreases the detected IR diffraction intensity and thus makes the same pattern show different IR structural colors. The finger can also selectively interact with different gratings by adjusting its position because gratings with different periods correspond to different incident angle ranges for the generation of IR structural colors. The interaction is flexible to the hand-interface distance ranging from a few centimeters to tens of centimeters. By taking advantage of the selective interaction between the grating structures and the finger, the dynamic and intelligent display of different color patterns can be generated. The generated IR structural colors can also be used as signals for the remote control of a robot vehicle. We demonstrated the use of only one finger to generate different IR structural colors and command a robot vehicle to perform complex actions in real-time and in the dark. The findings in this work provide an alternative approach for noncontact HMI with increased flexibilities and higher robustness both in the visible light condition and in the dark, which may further broaden the applications of HMI.

## Methods

**Fabrication of metallic gratings**. Metallic gratings with different structural parameters (periods and duty cycles) were fabricated by photolithography and thermal deposition. Photoresist solution (1 mL, RZJ-304, Suzhou Ruihong Electronic Chemical Co., Ltd.) was first dropped onto a piece of silicon (Si) wafer. The photoresist solution was spin-coated at the speed of 1000 r/min using a spin-coater (WS-650-23B, MYCRO ASIA PTE LTD.) for 1 min. The photoresist was then cured at 100 °C for 4 min. The Si wafer with cured photoresist was placed on the working table of a Mask Aligner (URE-2000/35, Institute of Optics and electronics, Chinese Academy of Sciences). Photomask with designed patterns of gratings was covered over the cured photoresist and then the cured photoresist was exposed to UV light for 15 s. The sample after UV exposure was developed by the positive developer (RZJ-3038, Suzhou Ruihong Electronic Chemical Co., Ltd.) for 15 s. Aluminum (Al) film with a thickness of 200 nm was deposited on the fabricated gratings by thermal evaporation apparatus (JSD400, Jiashuo Vacuum Technology Co., Ltd., China).

**Fabrication of grating arrays by soft lithography**. The fabrication process is shown in Supplementary Fig. 13. PDMS precursor was prepared by mixing the silicone elastomer base (Sylgard 184A, Dow Corning Corporation) and the curing agent (Sylgard 184B, Dow Corning Corporation) with the mass ratio of 10:1. The PDMS precursor was poured on the grating arrays fabricated on Si wafer and then cured at 70 °C for 3 h. After curing, the PDMS sample was peeled off from the Si wafer and the inverse structure of the gratings was imprinted on the surface of PDMS. The PDMS sample was then used to fabricate grating arrays on Si substrate. Prepolymer solution (1-methoxy-2-propyl acetate-based, 1 mL) was dropped onto a piece of Si wafer and was spin-coated at the speed of 500 r/min for 1 min. The PDMS stamp was then imprinted onto the prepolymer film to replicate the grating structures on the film. After curing for 4 min at 100 °C, the PDMS stamp was peeled off and the grating structures were generated on the Si substrate. The PDMS stamp itself can be used as a flexible grating as well. Gold (Au) film with a thickness of 100 nm was deposited on the PDMS stamp and the polymer gratings on Si substrate by thermal evaporation apparatus to enhance the diffraction efficiency of gratings.

**Experimental setup**. For the generation of IR structural colors, the sample of gratings was placed on a horizontal working table. The IR detector (FLIR T620) was fixed on a tripod and was placed about 30 cm away from the gratings. In the experiment, the lens of the IR detector was set at ~10° from the axis perpendicular to the center of the gratings (Supplementary Fig. 2). The hand was placed close to

the sample to cover the incident angle range of 11°–90° for the generation of the color palette. The finger was placed at different positions to generate different IR structural colors.

## Data availability

All relevant data are included in the paper and the Supplementary Information files.

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

## Acknowledgements

The authors want to thank Prof. Jingquan Liu and Chunpeng Jiang for their helpful suggestions and discussions. Thank Zhihui Lei, Yingyue Zhang, and Boning Shi for the help with the experiments. The authors acknowledge the Center of Hydrogen Science of Shanghai Jiao Tong University, the Instrumental Analysis Center, and the Zhiyuan Innovative Research Center of Shanghai Jiao Tong University for their support. This research was supported by the National Natural Science Foundation of China (Grant nos. 51521004, T.D.; 51973109, C.S.; 51873105, P.T.), the Innovation Program of Shanghai Municipal Education Commission (Grant no. 2019-01-07-00-02-E00069, T.D.).

## Author contributions

W.S., T.D., and S.A. conceived and planned this study. S.A., H.Z., and C.G. carried out experimental work. All authors (S.A., H.Z., C.G., B.F., C.S., P.T., W.S., and T.D.) discussed the results and contributed to the writing of the paper.

## Competing interests

The authors declare no competing interest.
