## [Peer Review File · Nature Communications]

Noncontact human-machine interaction based on hand-responsive infrared structural colorReviewers' comments:

Reviewer #1 (Remarks to the Author):

In this work, the author reported an IR structural color-based HMI for user-interactive touchless display and real-time control of a robot vehicle. Metallic gratings are used as the human-machine interface to respond to IR radiation from hand and the generated signals are visualized as different IR structural colors. Thus, the hand was creatively employed as an infrared (IR) light source for noncontact HMI. Even in the darkness, the interaction is flexible to the hand-interface distance ranging from a few centimeters to tens of centimeters and can be used in low lighting conditions. This investigation is certainly interesting and novel. As a result, several concerns in the manuscript need to be addressed before consideration of acceptance, as follows:

1. Compared with the HMI composed of the infrared thermal imager, what are the advantages of this report?
2. What is the meaning of IR structural colors? The HMI in this work only needs to recognize the action or the switch value. Why do we need to add the recognition value of color?
3. One advantage of this noncontact HMI is that it can be used in low lighting conditions or in the dark, and the interaction principle is the specific interaction between hand and different gratings, but in a dark environment, how could the user determine the relative position between the fingers and the HMI?
4. How does hand perspiration show an impact on the noncontact human-machine interaction?
5. How does the environmental temperature variation show an impact on the device performance?
6. In the Introduction, when mentioning pressure-based human-machine interfaces, a classic work could be credited (ACS Nano 2015, 9, 105).
7. It is suggested to polish the English writing and reduce the typos. The reference format needs to be carefully revised to fit the journal requirement. Some of the current reference formats are not consistent with each other, for example, some article titles are capitalized, some are not.

Reviewer #2 (Remarks to the Author):

This manuscript describes non-contact mode human-machine interface platform in which as a simple demonstration, the motion of a toy vehicle is controlled by hand gestures. The method is based on IR-responsive structural color arising from a periodically patterned metallic grating. More specifically, IR emission from a hand or a finger was interacted with the periodic metal structure, giving rise to the

unique structural color in IR (constructive interference between the pattern and IR from the hand). The structural color depends upon not only the characteristics of a pattern such as periodicity and duty cycle but also incident angle of the input IR. By carefully controlling those materials and geometric factors, the authors successfully demonstrate that a finger motion with respect to a periodic pattern is recognized by the different structural color. The structural color in IR regime was converted to visible one via a microprocessor. In addition, the motion-dependent structural color was transferred to the motion switches of a toy vehicle, making the vehicle move around by the programmed hand motion. The authors claim that the system was long-distance responsive from a few centimeters to tens of centimeters, fast responsive, and reliable. In addition, by utilizing IR arising from human body, the motion interaction was done in dark condition, broadening its suitability.

The motivation of the work for human-machine interaction is clearly accomplished by the interesting approach based on structural color ascribed to the interaction between a periodic pattern and IR from human body. Considering that most of works have dealt with visible structural colors for direct visualization of stimuli without additional signal converting technique, the work utilizing IR structural color seems unique. However, at the same time, it is a weakness of the work since the system requires a microprocessor to convert IR structural color to either visible one or sensor signal for triggering the vehicle motion. In this sense, the work should be carefully compared with IR detection and sensing technologies with which human natural IR is readily detected (e.g. IR thermal camera). IR from a finger can be detected by a conventional IR detector as a function of the finger distance and position, and the information of the IR detector can be transferred to either visible display or motion switch of a toy vehicle similar to what the current work demonstrated based on IR structural color. In other words, the various demonstrations shown in the work based on IR structural color could be done with a conventional IR detector. This issue should be clarified.

A possible problem of the current approach is that carefully designed pattern structure should be prepared for large area for facile detection of a finger motion. The fabrication of photolithographic patterns seems cost ineffective. In addition, the rigid metallic pattern is hardly altered. Another issue is the interference of an IR from a finger with various IR sources from environment with different IR wavelengths (heat sources). The working principle is based on IR intensity variation depending upon incident angle of IR source from a finger, which seems continuously varied with the incident angle (Figure 3f). First of all, theoretical analysis of this behavior should be provided to confirm the experimental results. Next, the authors should think about how to develop so called "reliable motion sensing" for a specific motion with such an analogue type of intensity variation. Broad specification of R1, R2, and R3 seems quite vague. The device reliability, detection speed, and resolution and so on should be more quantitatively analyzed. Most critically, the work based on IR structural color should be clearly distinguished from conventional non-contact IR detection system.

Comments from Reviewer # 1

General comment:

In this work, the author reported an IR structural color-based HMI for user-interactive touchless display and real-time control of a robot vehicle. Metallic gratings are used as the human-machine interface to respond to IR radiation from hand and the generated signals are visualized as different IR structural colors. Thus, the hand was creatively employed as an infrared (IR) light source for noncontact HMI. Even in the darkness, the interaction is flexible to the hand- interface distance ranging from a few centimeters to tens of centimeters and can be used in low lighting conditions. This investigation is certainly interesting and novel. As a result, several concerns in the manuscript need to be addressed before consideration of acceptance, as follows:

Response: Thank the reviewer for the encouraging comments. Based on the comments from the reviewer, we have thoroughly revised the manuscript to address the reviewer's concerns. Following are the point-by-point responses to the detailed comments from the reviewer.

1. Compared with the HMI composed of the infrared thermal imager, what are the advantages of this report?

Response: We want to thank the reviewer for the comment. In the HMI composed of the infrared (IR) thermal imager, IR thermal imager or IR camera is widely used to directly capture the IR images of gestures. The captured gestures are recognized as specific commands for HMI, which can be referred as direct gesture-based HMI. Direct gesture-based HMI can be divided into four stages and each stage has its challenge (Fig. R1-1, left column): (1) Capture of IR images of gestures. The hand is a complex articulated object with more than 20 degrees of freedom. The gestures are thus complex and IR images of gestures contain a large amount of information, which makes real-time HMI challenging. (2) Gesture segmentation from the background. In this stage, the complex background will interfere the segmentation of gestures and thus affect the recognition performance. (3) Gesture modeling and feature extraction. Different gestures are distinguished from each other by their unique features. The self-occlusion, which means the block or overlap of fingers by other fingers or other parts of the hand, may interfere the extraction of features and thus affect the recognition performance. (4) Gesture recognition as specific command. In this stage, a classifier recognizes gestures as specific commands based on the incoming features. Due to the interference of complex background and self-occlusion in stage 2 and 3, the gesture recognition process is highly dependent on algorithms and the complex algorithms are needed for the recognition process. To overcome these challenges in direct gesture-based HMI, an alternative approach may provide different path to the effective and accurate gesture recognition.

Different from the direct capture and recognition of complex gestures by IR camera, here we used gratings to serve as the interface for the selective interaction with the IR light from fingers, and to convert complex gestures into easily recognizable patterns of IR structural colors (Fig. R1-1, right column). Such selective interaction and conversion process helps minimize the interference of background and the self-occlusion of the fingers. The generated simple and recognizable patterns of IR structural colors also generate much less data to be processed and require less complex algorithms in the analysis process, both of which help simplify the signal recognition process. Following are the more detailed description of comparison that shows the advantages offered by the IR structural color-based HMI presented in this study:

Fig. R1-1. The comparison between HMI composed of the IR imager with direct imaging of gesture and IR structural color-based HMI.

(1) Amount of information captured for processing

In direct gesture-based HMI, real-time recognition is a challenge because gestures are relatively complex and irregular, which makes the captured images of gestures contain large amount of information. The processing of these images is time-consuming. Using the grating array as the human-machine interface, the complex gestures are converted into simple and recognizable color

patterns. The simplification of the recognized patterns can reduce the amount of information of each image, which is beneficial for real-time recognition. Here we use information entropy to quantitatively compare the amount of information between IR images of gestures directly captured by an IR imager and the IR color patterns generated from the hand-grating interaction. The smaller entropy means the less amount of information. In the original manuscript, we demonstrate that we can use individual finger as the IR light source to interact with the grating array and generate different patterns of IR structural colors by vertically moving the finger to the incident angle ranges of different gratings (G_g , G_y , and G_r) (Fig. R1-2a). Here we compared the entropy of two processes (IR structural color-based imaging and direct IR imaging of gestures) by using one finger. As shown in the original manuscript and also in Fig. R1-2a, we can generate three different IR color patterns (p_1 , p_2 , and p_3) by moving the index finger in the IR structural color-based imaging process. To compare with the direct IR imaging of gestures, we also directly captured the IR images of the same gesture at the positions where these three IR color patterns were generated. The directly captured IR images of the gesture at different positions, which are named as g_1 , g_2 , g_3 , are shown in Fig. R1-2b. We then calculated the information entropy of these IR images respectively. As shown in Fig. R1-2c (left), the information entropy of directly captured IR images of gestures (1.42 bits/pixel on average) is much larger than that of IR color patterns (0.16 bits/pixel on average). In the original manuscript, we also demonstrate that we can use the whole hand as the IR light source to interact with the grating array and generate different patterns of IR structural colors. Here we further compared the entropy of two processes (IR structural color-based imaging and direct IR imaging of gestures) by using the whole hand as well. The comparison process is the same as the above comparison process, with the generated IR color patterns named as p_4 , p_5 , and p_6 and the corresponding gestures named as g_4 , g_5 , and g_6 respectively. As shown in Fig. R1-2c (right), the average information entropy of IR color patterns generated by the gesture of the whole hand is 0.16 bits/pixel, also much less than the average information entropy of gesture of the whole hand (1.48 bits/pixel). The small information entropy means that IR color patterns contain much less amount of information in the signal processing, which is beneficial for real-time recognition.

Fig. R1-2. The comparison of information entropy of captured IR images. **a** IR color patterns (p1, p2, and p3) generated by vertically moving the finger to the incident angle ranges of different gratings (Gg, Gy, and Gr). **b** Directly captured IR images of corresponding gestures (g1, g2, and g3) that generate the IR color patterns (p1, p2, and p3) in Fig. R1-2a. **c** The calculated entropy of the IR images in Fig. R1-2a and b (left two columns) and the calculated entropy of IR images by using the whole hand as the IR light source (right two columns). p4, p5, and p6 represent the generated IR color patterns by using the whole hand as the IR light source. g4, g5, and g6 represent the directly captured IR images of corresponding gestures that generate the IR color patterns (p4, p5, and p6).

We added Fig. R1-2 as Supplementary Figure 17 in the revised Supplementary Information and added the following statement in the revised Supplementary Information as Supplementary Text, Section 5 to describe the detailed calculation process of information entropy:

“Section 5. Calculation of two-dimensional (2D) information entropy of captured IR images

2D information entropy is defined by the grayscale distribution of an image, which can characterize the amount of data in the image. For each pixel of the image, the grayscale of the pixel and the average grayscale of its neighborhood are first calculated, which are i and j respectively. The grayscale of the pixel (i) and the average grayscale of the neighborhood (j) form a pair. The probability of each pair is then calculated to be p_{ij} . The 2D entropy can be calculated by the formula:

$$E = - \sum_{i=0}^{255} \sum_{j=0}^{255} p_{ij} \log_2 p_{ij} \quad (18)''$$

We also added a section “Comparison between direct gesture-based HMI and IR structural color-based HMI” in the revised Supplementary Information as Supplementary Text, Section 6 to address the reviewers’ concern and added the following statement in this section to show the difference of the amount of information captured:

“IR camera is also widely used to directly capture gestures. The captured gestures are recognized as specific commands for HMI, which can be referred as direct gesture-based HMI. Compared with direct gesture-based HMI, the conversion of complex gestures into simplified IR color patterns can decrease the amount of information in the captured image and thus is beneficial for real-time recognition. The amount of information of an image can be characterized by information entropy (Supplementary Text, Section 5). The smaller entropy means the less amount of information. In Fig. 3 we demonstrate that we can use individual finger as the IR light source to interact with the grating array and generate different patterns of IR structural colors by vertically moving the finger to the incident angle ranges of different gratings (Gr, Gy, and Gg). Here we quantitatively compared the entropy of two processes (IR structural color-based imaging and direct IR imaging of gestures) by using one finger. As shown in Supplementary Figure 17a, we can generate three different IR color patterns (p1, p2, and p3) by moving the index finger in the IR structural color-based imaging process. To compare with the direct IR imaging of gestures, we also directly captured the IR images of the same gesture at the positions where these three IR color patterns were generated. The directly captured IR images of the gesture at different positions, which are named as g1, g2, g3, are shown in Supplementary Figure 17b. We calculated the information entropy of these IR images respectively. As shown in Supplementary Figure 17c (left), the information entropy of directly captured IR images of gestures (1.42 bits/pixel on average) is much larger than that of IR color patterns (0.16 bits/pixel on average). In Supplementary Figure 7, we also demonstrate that we can use the whole hand as the IR light source to interact with the grating array and generate different patterns of IR structural colors. Here we compared the entropy of two processes (IR structural color-based imaging and direct IR imaging of gestures) by using the whole hand as well. The comparison process is the same as the above comparison process, with the generated IR color patterns named as p4, p5, and p6 and the corresponding gestures named as g4, g5, and g6 respectively. As shown in Supplementary Figure 17c (right), the average information entropy of IR color patterns generated by the gesture of the whole hand is 0.16 bits/pixel, also much less than the average information entropy of gesture of the whole hand (1.48 bits/pixel). The smaller information entropy means that IR color patterns contain less amount of information, which is beneficial for real-time recognition.”

(2) Interference of background

In direct gesture-based HMI, the IR imager captures the IR image of the whole hand and the background. The complex background and the other parts of the hand will interfere the recognition of the gesture (Wang, M. *et al.* Gesture recognition using a bioinspired learning architecture that integrates visual data with somatosensory data from stretchable sensors. *Nat. Electron.* **3**, 563-570 (2020). Chakraborty, B. K. *et al.* Review of constraints on vision-based gesture recognition for human–computer interaction. *IET Comput. Vis.* **12**, 3-15 (2018)). The direct gesture-based HMI process thus requires the segmentation of gesture from the background (Fig. R1-1).

In comparison, the IR structural color-based HMI uses the grating array as the interface to selectively interact with fingers, which minimizes the interference of background and other parts of the hand. As discussed in the original manuscript, the relationship between incident angle (θ_{in}) and diffraction angle (θ_d) satisfies the diffraction equation:

$$\sin\theta_m = \sin\theta_d - \frac{m\lambda}{\Lambda} \quad (\text{R1-1})$$

where m is the diffraction order and is a nonzero integer, λ is the wavelength of the light, and Λ is the grating period. -1st diffraction efficiency ($m = -1$) in general is higher than diffraction efficiency of other orders, so we focus on -1st diffraction in this work. With the camera fixed at 10° from the axis perpendicular to the center of the gratings, the detected diffraction angle is $\theta_d = 10^\circ$, which means that only the diffracted light with the angle of 10° can be captured and detected by the camera while diffracted light with other angles cannot be captured by the camera. Based on the detected diffraction angle and the grating period, we can calculate the corresponding incident angle according to Equation R1-1, which is an angle range corresponding to the range of detectable IR light wavelength by the camera (7.5 μm ~ 14 μm). Equation R1-1 shows that not all the IR radiation (from hand and from the background) can be detected by the IR camera. Only the IR radiation with incident angle satisfying Equation R1-1 can interact with gratings and then the diffracted light can be detected by the IR camera, which minimizes the interference of IR radiation from other parts of the hand and the complex background. Besides the theoretical analysis, we further ran an experiment to demonstrate the reduction of the interference from the IR radiation emitted by the background and other parts of the hand due to the selective interaction between hand and gratings (Fig. R1-3). The grating used in this experiment is Gr with the period of 25 μm in Fig. 3e of the original manuscript. For the detected diffraction angle ($\theta_d = 10^\circ$), we can calculate the corresponding incident angle range to be 28° ~ 47°.

When we placed the index finger in the calculated incident angle range, the IR radiation from the finger can be diffracted by the grating with the diffraction angle of $\theta_d = 10^\circ$ (Fig. R1-3a). Apparently, IR radiation from the other part of the hand (the palm) and the background can also interact with the grating. To confirm whether the IR radiation from other parts of the hand can interfere the generation of IR color patterns, we used an Al foil with low emissivity (~0.01) to cover the hand except the index finger, which limits the IR radiation from other parts of the hand (Fig. R1-3b). We compared the IR diffraction intensity of the grating with the hand not covered and covered with the Al foil. As shown in Fig. R1-3e, the cover of the hand with Al foil does not change the IR diffraction intensity of the grating, which means that the IR radiation from the other parts of the hand does not interfere the selective interaction between the index finger and the grating. We also placed a heat source with the temperature of ~80 °C at different positions (Fig. R1-3c and d) and compared the IR diffraction intensity of the grating before and after adding the heat source. As shown in Fig. R1-3e, the extra heat source does not change the IR diffraction intensity of the grating, which means that only the IR radiation with the calculated incident angle can selectively interact with gratings and detected by the IR camera. Other IR radiation does not interfere the generation of IR structural colors. The selective interaction between finger and gratings minimizes the interference of complex background and other parts of the hand on the recognition of gestures and commands.

Fig. R1-3. The reduction of interference from background and other parts of the hand due to the selective finger-grating interaction. **a** Placing the index finger in the calculated incident angle range. **b** Covering the palm with the Al foil. **c** Placing a heat source with the temperature of $\sim 80^\circ\text{C}$ above the finger. **d** Placing a heat source with the temperature of $\sim 80^\circ\text{C}$ on the other side of gratings. **e** The IR diffraction intensity of the gratings under the above four conditions. Error bars represent the standard deviation of the mean.

We added Fig. R1-3 as Supplementary Figure 18 in the revised Supplementary Information and added the following statement in the section of “Comparison between direct gesture-based HMI

and IR structural color-based HMI” in the Supplementary Information to address the reviewer’s concern:

“In direct gesture-based HMI, the IR camera captures the IR image of the whole hand and the background. The complex background and the other parts of the hand will interfere the recognition of the gesture. In comparison, this IR structural color-based HMI uses the grating array as the interface to selectively interact with fingers, which minimizes the interference of background and other parts of the hand. To demonstrate such minimized interference due to the selective interaction between hand and gratings, we used Gr with the period of 25 μm (Fig. 3e) to interact with the hand (Supplementary Figure 18). For the detected diffraction angle ($\theta_d = 10^\circ$), the corresponding incident angle range is calculated to be $28^\circ \sim 47^\circ$. When we placed the index finger in the calculated incident angle range, the IR radiation from the finger can be diffracted by the grating with the diffraction angle of $\theta_d = 10^\circ$ (Supplementary Figure 18a). Apparently, IR radiation from the other parts of the hand (the palm) and the background can also interact with the grating. To confirm whether the IR radiation from other parts of the hand can interfere the generation of IR diffraction patterns, we used an Al foil with low emissivity (~ 0.01) to cover the hand except the index finger, which limits the IR radiation from other parts of the hand (Supplementary Figure 18b). We compared the IR diffraction intensity of the grating with the hand not covered and covered with the Al foil. As shown in Supplementary Figure 18e, the cover of the hand with Al foil does not change the IR diffraction intensity of the grating, which means that the IR radiation from the other parts of the hand does not interfere the selective interaction between the index finger and the grating. We also placed a heat source with the temperature of $\sim 80^\circ\text{C}$ at different positions (Supplementary Figure 18c and 18d) and compared the IR diffraction intensity of the grating before and after adding the heat source. As shown in Supplementary Figure 18e, the extra heat source does not change the IR diffraction intensity of the grating, which means that only the IR radiation with the calculated incident angle can selectively interact with gratings and detected by the IR camera. Other IR radiation does not interfere the generation of IR structural colors. The selective interaction between hand and gratings minimizes the interference of complex background and other parts of the hand on the recognition of gestures and commands.”

We also added the following statement in the revised manuscript (Page 11, Paragraph 1, Line 2) for the theoretical analysis of the selective interaction between finger and gratings:

“In the interaction between finger and the grating, the other parts of the hand and the background also emit IR radiation and interact with the grating. Due to the fix of the detected diffraction angle, not all the IR radiation (from hand and from the background) can be detected by the IR camera after the interaction with gratings. Only the IR radiation with incident angle satisfying diffraction equation (Equation 1) can interact with gratings and then the diffracted light is detected by the IR camera. The selective interaction between finger and grating minimizes the interference of IR radiation from other parts of the hand and the complex background on the generation of IR structural colors.”

(3) Interference of self-occlusion

Direct gesture-based HMI generally involves gestures composed of multiple fingers. In this case, self-occlusion is hard to avoid, both in the direct imaging using visible imagers or IR imagers, because the hand is a complex articulated object with more than 20 degrees of freedom (Ge, L. et al. Real-time 3D hand pose estimation with 3D convolutional neural networks. *IEEE Trans. Pattern Anal. Mach. Intell.* 41, 956-970 (2019). Chakraborty, B. K. et al. Review of constraints

on vision-based gesture recognition for human–computer interaction. *IET Comput. Vis.* **12**, 3-15 (2018)). The self-occlusion of gestures will interfere the extraction of features from gestures and thus affect the recognition performance of HMI. In the IR structural color-based process presented in this work, the highly selective interaction between individual finger and the corresponding grating minimizes the impact of self-occlusion, since grating only interacts with the corresponding finger, while other fingers or other parts of the hand will not interfere such interaction, as demonstrated in Fig. R1-3. We further ran an experiment to use two fingers in the two processes (direct IR imaging of gestures and the IR structural color-based imaging) to compare the effect of self-occlusion on the generated IR images. This experiment involves three gestures (g7, g8, g9 in Fig. R1-4a-c) for the direct imaging process. For comparison, in the IR structural color-based process, we used two gratings (Gr and Gg in Fig. 3e in the original manuscript) to interact with these three same gestures. As shown in Fig. R1-4a, we first used the extending middle finger and index finger as the gesture (g7). We then bent the middle finger to 90° to generate a new gesture (g8, Fig. R1-4b). We further bent the middle finger to the palm and generated another gesture (g9, Fig. R1-4c). For the first process that involves the direct capturing of the IR images of the three gestures, we captured the three gestures using IR camera from two different angles (front view and side view). From the captured IR images of gestures (Fig. R1-4d-i) we can find that self-occlusion is hard to avoid and interferes the extraction of features from gestures. For example, in the IR image of g8 captured from the front view (Fig. R1-4e), the bending part of the middle finger is overlapped in this angle (white rectangle), which makes the feature of bending the middle finger unrecognizable in the captured IR image and makes the differentiation of IR images of g8 (Fig. R1-4e) and g9 (Fig. R1-4f) a little challenging. When the IR camera captured gestures from the side view, the bending of the middle finger is recognizable (Fig. R1-4h). In this case, however, it is hard to differentiate the captured IR image of g7 (Fig. R1-4g) from that of g9 (Fig. R1-4i) due to the block of the middle finger by the index finger (blue rectangle). The self-occlusion is thus hard to avoid in the direct gesture-based HMI, which interferes the extraction of features from gestures and the recognition performance of gestures. In comparison, for the IR structural color-based process, we captured the IR color patterns using the same three gestures. We first placed the gesture g7 in the incident angle range of Gr to generate the IR diffraction pattern of “red” on Gr (p7, Fig. R1-4j). We then bent the middle finger to 90° to generate a gesture g8. The bending part of the middle finger covered the incident angle range of Gg and generated the IR diffraction pattern of “green” on Gg (p8, Fig. R1-4k). Meanwhile, the covered incident angle range of Gr decreased (Fig. R1-4b) and thus the “red” on Gr turned to “yellow” (Fig. R1-4k). With the further bending of the middle finger to the palm (Fig. R1-4c), the middle finger cannot cover the incident angle range of Gg and thus “green” turned off in this case (p9, Fig. R1-4l). The selective interaction between fingers and gratings converted features of gestures into patterns of IR structural colors with minimized interference of self-occlusion. As shown in Fig. R1-4j-l, the three IR color patterns can be clearly differentiated based on the color of Gr and Gg. The minimized interference of self-occlusion can enhance the recognition performance of HMI.

Fig. R1-4. The characterization of the interference of self-occlusion on the recognition of features. **a-c** Schematic illustration of three gestures (g7-g9). **d-f** Directly captured IR images of gestures from the front view. **g-i** Directly captured IR images of gestures from the side view. The white rectangle and the blue rectangle show the self-occlusion of gestures, which interferes the extraction of features. **j-l** Generated IR color patterns (p7-p9) by corresponding gestures.

We added Fig. R1-4 as Supplementary Figure 19 in the revised Supplementary Information and added the following statement in the section of “Comparison between direct gesture-based HMI and IR structural color-based HMI” in the Supplementary Information to address the reviewer’s concern:

“Direct gesture-based HMI generally involves gestures composed of multiple fingers. In this case, self-occlusion is hard to avoid because the hand is a complex articulated object with more than 20 degrees of freedom. The self-occlusion of gestures will interfere the extraction of features from gestures and thus affect the recognition performance of HMI. In the IR structural color-based process, the highly selective interaction between individual finger and the corresponding grating minimizes the impact of self-occlusion, since grating only interacts with the corresponding finger, while other fingers or other parts of the hand will not interfere such interaction. We further ran an experiment to use two fingers in the two processes (direct IR imaging of gestures and the IR structural color-based imaging) to compare the effect of self-occlusion on the generated IR images. This experiment involves three gestures (g7, g8, g9 in Supplementary Figure 19a-c) for the direct imaging process. For comparison, in the IR structural color-based process, we used two gratings (Gr and Gg in Fig. 3e) to interact with these three same gestures. As shown in Supplementary Figure 19a, we first used the extending middle finger and index finger as the gesture (g7). We then bent the middle finger to 90° to generate a new gesture (g8, Supplementary Figure 19b). We further bent the middle finger to the palm and generated another gesture (g9, Supplementary Figure 19c). For the first process that involves the direct capturing of the IR images of the three gestures, we captured the three gestures using IR camera from two different angles (front view and side view). From the directly captured IR images of gestures (Supplementary Figure 19d-i) we can find that self-occlusion is hard to avoid

and interferes the extraction of gesture features. For example, in the IR image of g8 captured from the front view (Supplementary Figure 19e), the bending part of the middle finger is overlapped in this angle (white rectangle), which makes the feature of bending the middle finger unrecognizable in the captured IR image and makes the differentiation of IR images of g8 (Supplementary Figure 19e) and g9 (Supplementary Figure 19f) a little challenging. When the IR camera captured gestures from the side view, the bending of the middle finger is recognizable (Supplementary Figure 19h). In this case, however, it is hard to differentiate the captured IR image of g7 (Supplementary Figure 19g) from that of g9 (Supplementary Figure 19i) due to the block of the middle finger by the index finger (blue rectangle). The self-occlusion is thus hard to avoid in the direct gesture-based HMI, which interferes the extraction of features from gestures and the recognition performance of gestures. In comparison, for the IR structural color-based process, we captured the IR color patterns using the same three gestures. We first placed the gesture g7 in the incident angle range of Gr to generate the IR diffraction pattern of “red” on Gr (p7, Supplementary Figure 19j). We then bent the middle finger to 90° to generate gesture g8. The bending part of the middle finger covered the incident angle range of Gg and generated the IR diffraction pattern of “green” on Gg (p8, Supplementary Figure 19k). Meanwhile, the covered incident angle range of Gr decreased (Supplementary Figure 19b) and thus the “red” on Gr turned to “yellow” (Supplementary Figure 19k). With the further bending of the middle finger to the palm (Supplementary Figure 19c), the middle finger cannot cover the incident angle range of Gg and thus “green” turned off in this case (p9, Supplementary Figure 19l). The selective interaction between fingers and gratings converted features of gestures into patterns of IR structural colors with minimized interference of self-occlusion. As shown in Supplementary Figure 19j-l, the three IR color patterns can be clearly differentiated based on the color of Gr and Gg. The minimized interference of self-occlusion can greatly enhance the recognition performance of HMI.”

(4) Simplified algorithm

Due to the complexity of gesture and interference of background and self-occlusion, direct gesture-based HMI includes four stages (Fig. R1-1). Except the capture of gestures by IR camera, other three stages, including gesture segmentation from the background, gesture modeling and feature extraction, and gesture recognition as specific commands are processed by algorithms. The recognition performance of gestures is highly dependent on algorithms. Much effort in the direct gesture-based HMI thus focuses on the development of algorithms.

In comparison, our work uses a grating array to interact with hand. The feature of each gesture is directly converted into easily recognizable patterns of IR structural colors. The use of grating array thus simplifies the algorithm to directly recognize features of images, with no need of gesture segmentation from the background and the hand modeling and feature extraction. As discussed in original Supplementary Information, Section 3, the algorithm just needs to extract the color of each grating and then the corresponding command will be recognized. The simplified algorithm is beneficial for real-time HMI without the need of relatively complex hardware and software for the required algorithms used in the direct gesture-based HMI.

We added the following statement in the section of “Comparison between direct gesture-based HMI and IR structural color-based HMI” in the Supplementary Information to address the reviewer’s concern:

“Direct gesture-based HMI usually includes four stages. Except the capture of gestures by IR camera, other three stages, including gesture segmentation from the background, gesture

modeling and feature extraction, and gesture recognition as specific commands are all processed by algorithms. The recognition performance is thus highly dependent on the algorithms. In comparison, the use of grating array to selectively interact with gestures minimizes the interference of complex background and self-occlusion. The feature of each gesture is directly converted into easily recognizable patterns of IR structural colors. This IR structural color-based HMI thus simplifies the algorithm to recognize features of images, with no need of gesture segmentation from the background and the hand modeling and feature extraction. The simplified algorithm is beneficial for real-time HMI without complex hardware and software.”

We also added the following statement in the revised manuscript (Page 20, Paragraph 2, Line 13) to summarize the comparison result between direct gesture-based HMI and IR structural color-based HMI:

“Comparison between direct-gesture based HMI and IR structural color-based HMI. IR camera is widely used to directly capture the IR images of gestures. The captured gestures are recognized as specific commands for HMI, which can be referred as direct gesture-based HMI. Different from the direct capture and recognition of complex gestures by IR camera, in this work we use gratings to serve as the interface for the selective interaction with the IR light from fingers, and to convert complex gestures into easily recognizable patterns of IR structural colors. We further compare gesture-based HMI and IR structural color-based HMI from four aspects: (1) Amount of information captured for processing. The hand is a complex articulated object with more than 20 degrees of freedom. The gestures are thus complex and IR images of gestures contain a large amount of information, which makes real-time HMI challenging. In contrast, the IR structural color patterns contain much less amount of information, which is beneficial for real-time recognition (Supplementary Text, Section 5 and Supplementary Figure 17). (2) Interference of background. In direct gesture-based HMI, the complex background will interfere the segmentation of gestures and thus affect the recognition performance. In comparison, this IR structural color-based HMI uses the grating array as the interface to selectively interact with fingers, which minimizes the interference of background and other parts of the hand (Supplementary Figure 18). (3) Interference of self-occlusion. In direct gesture-based HMI, different gestures are distinguished from each other by their unique features. The self-occlusion, which means the block or overlap of fingers by other fingers or other parts of the hand, is hard to avoid and interferes the extraction of features. In IR structural color-based HMI, the highly selective interaction between individual finger and the corresponding grating minimizes the impact of self-occlusion, since grating only interacts with the corresponding finger at specific position, while other fingers or other parts of the hand will not interfere such interaction (Supplementary Figure 19). (4) Algorithm. Due to the interference of complex background and self-occlusion, the gesture recognition process is highly dependent on algorithms and complex algorithms are needed for the recognition process. In comparison, IR structural color-based HMI uses a grating array to interact with hand. The feature of each gesture is directly converted into easily recognizable patterns of IR structural colors. The use of grating array thus simplifies the algorithm to directly recognize features of images, with no need of gesture segmentation from the background and the hand modeling and feature extraction. In summary, the selective interaction between hand and gratings in IR structural color-based HMI helps minimize the interference of background and the self-occlusion of the hand. The generated simple and recognizable patterns of IR structural colors also generate much less data to be processed and require less complex algorithms in the analysis process, both of which help simplify the signal recognition process. The detailed comparison result is shown in Supplementary Text, Section 6.”

2. What is the meaning of IR structural colors? The HMI in this work only needs to recognize the action or the switch value. Why do we need to add the recognition value of color?

Response: We want to thank the reviewer for the comment. In this study, we define the IR structural color as the intensity of IR radiation diffracted by gratings. The IR radiation from hand/finger is diffracted by different gratings with different intensities, which are characterized and visualized by different IR structural colors on the IR imagers.

We agree with the reviewer that the human-robot vehicle interaction in this work only needs to recognize the action or the switch value such as the switch value of electrical signals, which are directly generated by IR radiation with different intensities. The reason to have the recognition value of color is two folds: (1) The structural color has the advantage of direct visualization of stimuli. The visible structural color is widely used in HMI, for example, user-interactive color display (Kang, H. S. et al. 3D touchless multiorder reflection structural color sensing display. *Sci. Adv.* **6**, eabb5769 (2020)). In this work, we convert IR diffraction intensities into IR structural colors, which can be used for the direct visualization of the interaction between hand/finger and different gratings, similar to visible structural colors. Compared with directly using the action or the switch value as the recognized signals, IR structural colors are more friendly for users to determine the relative position between finger and grating and judge whether the generated signal is correct. (2) The use of IR structural color for signal transfer also broadens the potential applications of IR radiation in HMI. Besides human-robot vehicle interaction, the IR structural color also enables user-interactive color display (Fig. 2d in the original manuscript). Combined with coding pattern (Quick Response code in Supplementary Figure 5c in the original Supplementary Information), the IR structural color also shows promise for information encryption and anti-counterfeiting.

We added the following statement in the revised manuscript (Page 4, Paragraph 1, Line 6) to address the reviewer's concern:

“The conversion of IR diffraction intensities into IR structural colors realizes the direct visualization of the interaction between hand and different gratings, which is friendly for users to determine the relative position between hand and grating and judge whether the generated signal is correct.”

We also revised the following statement in the revised manuscript (Page 4, Paragraph 1, Line 10) to address the reviewer's concern:

Original statement:

“We can place the hand close to the grating interface for user-interactive touchless display.”

Revised statement:

“We can place the hand close to the grating interface for user-interactive touchless display and multilevel information encryption/decryption.”

3. One advantage of this noncontact HMI is that it can be used in low lighting conditions or in the dark, and the interaction principle is the specific interaction between hand and different gratings, but in a dark environment, how could the user determine the relative position between the fingers and the HMI?

Response: Thank the reviewer for the comment. As discussed above, the interaction between finger and grating interface is visualized by IR structural colors. IR structural colors can thus be used for positioning. In a dark environment, we first move the finger to generate one pattern of IR structural color. Based on the generated IR structural color, we can determine the relative position between finger and grating array and then we can move the finger to generate other different patterns of IR structural colors.

The grating array can also be integrated into a wrist band and attached on the wrist to fix the relative position between fingers and gratings and thus enhance the operability of the HMI in the dark. As a proof-of-concept demonstration, we fabricated a grating array composed of two same gratings with period of $20\ \mu\text{m}$ and duty cycle of 50%. As shown in Fig. R1-5a, two gratings are attached on a wrist band and aligned with the middle finger and the ring finger respectively. The two gratings are invisible in the visible image of the wrist band so we use two squares with different colors to indicate the relative position of the two gratings. The grating corresponding to blue square is named as Gb and the grating corresponding to the violet square is named as Gv. With the bending of middle finger, the IR radiation from the middle finger will selectively interact with Gb. The covered incident angle range is adjustable by bending the finger with different degrees. Similarly, the ring finger can selectively interact with Gv by bending the finger. As shown in Fig. R1-5b and Movie R1, different IR color patterns are generated with the bending of different fingers. In the future, IR detectors may be also integrated into the wrist band to fix the relative position among fingers, grating array, and IR detectors, which can further enhance the reliability and portability of the HMI system.

Fig. R1-5. Integration of grating array into a wrist band. **a** Two gratings (blue grating (Gb) and violet grating (Gv)) are integrated into a wrist band, which are aligned with the middle finger and the ring finger respectively. **b** The generated IR color patterns with the bending of middle finger/ring finger.

We added Fig. R1-5 as Fig. 4e and Fig. 4f in the revised manuscript and added Movie R1 as Supplementary Movie 3 in the revised Supplementary Information. We also added the following statement in the revised manuscript (Page 19, Paragraph 2, Line 12) to address the reviewer's concern:

“In a dark environment, we first move the finger to generate one pattern of IR structural color. Based on the generated IR structural color, we can determine the relative position between finger and grating array and then move the finger to generate other patterns of IR structural colors. The grating array can also be integrated into a wrist band and attached on the wrist to fix the relative position between fingers and gratings and thus enhance the reliability and operability of HMI in the dark. As shown in Fig. 4e, two same gratings with period of $20\ \mu\text{m}$ and duty cycle of 50% are attached on a wrist band and aligned with the middle finger and the ring finger respectively. The two gratings are invisible in the visible image of the wrist band and we use two squares with

different colors to show the relative position of the gratings (Fig. 4e). The grating corresponding to blue square is named as Gb and the grating corresponding to the violet square is named as Gv. With the bending of middle finger, the IR radiation from the middle finger will selectively interact with Gb. The covered incident angle range is adjustable by bending the finger with different degrees. Similarly, the ring finger can selectively interact with Gv by bending the finger. Different IR color patterns are generated with the bending of different fingers (Fig. 4f and Supplementary Movie 3). In the future, IR detectors may also be integrated into the wrist band to fix the relative position among fingers, grating array, and IR detectors, which can further enhance the reliability and portability of the HMI system.”

4. How does hand perspiration show an impact on the noncontact human-machine interaction?

Response: We want to thank the reviewer for the comment. The hand perspiration will both generate a thin film of water on the hand and increase the humidity around the hand. For the generation of a water film, due to the low IR reflectivity of water, the hand and water film can act as a whole part with high emissivity, similar with the hand without perspiration. For the increase of humidity, due to the high IR transmissivity of H₂O vapor in the detected wavelength range and limited distance of humidity increase around hand (2 cm~3 cm to the hand, Kang, H. S. et al. 3D touchless multiorder reflection structural color sensing display. *Sci. Adv.* **6**, eabb5769 (2020)), IR radiation can transmit through the vapor with low attenuation. The hand perspiration thus has little effect on the noncontact HMI. We further ran an experiment to verify such analysis. We first used the dry hand to interact with gratings. We then exercised the hand to generate sweat and used the sweaty hand to interact with gratings. As shown in Fig. R1-6, with or without perspiration, the IR structural colors can both be generated.

Fig. R1-6. Effect of hand perspiration on the noncontact HMI. a The generated IR color patterns using dry hand as the IR light source. **b** The generated IR color patterns using sweaty hand as the IR light source. The insets are the images of dry hand and sweaty hand. We used allochroic silica gel to characterize the perspiration of hand. The silica gel is blue under dry environment and will change from blue to relatively pink/translucent under humid environment due to the absorption of water vapor.

We added Fig. R1-6 as Supplementary Figure 8 in the revised Supplementary Information and added the following statement in the revised manuscript (Page 15, Paragraph 2, Line 9) to address the reviewer’s concern:

“Hand perspiration has little effect on the interaction between hand and grating arrays due to the low IR reflectivity of H₂O film on the hand and the high IR transmissivity of H₂O vapor around the hand (Supplementary Figure 8).”

5. How does the environmental temperature variation show an impact on the device performance?

Response: We want to thank the reviewer for the comment. In this work, we ran experiments at the general indoor temperature of ~ 25 °C. To study the effect of environmental temperature on the device performance, we used grating Gr (Fig. 3e in the original manuscript) to interact with the index finger at different environmental temperature. As shown in Fig. R1-7, the dynamic IR structural colors can still be generated under different environmental temperature (T_e , 20 °C \sim 40 °C). When the environmental temperature is lower than the hand temperature (Fig. R1-7a-d), the interaction of hand and grating will increase the IR diffraction intensity from the grating. When the environmental temperature is higher than hand temperature ($T_e = 40.7$ °C in Fig. R1-7e), the IR radiation intensity of hand is lower than that of environmental background. In this case, the IR diffraction intensity of the grating is decreased when the hand is used to interact with the grating. The color bar is thus inversed to characterize the generated IR structural colors. Certainly when the environmental temperature is the same as the hand temperature, the IR radiation from hand cannot cause the change of IR diffraction intensity of gratings and thus cannot generate different IR structural colors. In this case, we can rub the hand to increase the hand temperature instantly, and thus increase the temperature differences between the hand and the environment. Fig. R1-7f shows that the temperature of hand can increase ~ 8 °C after a quick rubbing of the hand. This noncontact HMI can thus be used under different environmental temperature.

Fig. R1-7. Effect of environmental temperature (T_e) on the device performance. a-e The generated IR structural colors under different environmental temperature. f IR images of the index finger before and after rubbing with the other hand.

We added Fig. R1-7 as Supplementary Figure 9 in the revised Supplementary Information and added the following statement in the revised manuscript (Page 15, Paragraph 2, Line 11) to address the reviewer's concern:

“The interaction between hand and grating arrays is applicable for different environmental temperature (Supplementary Figure 9a-e). Specifically, when the environmental temperature is higher than hand temperature (Supplementary Figure 9e), the IR radiation intensity of hand is lower than that of environmental background. In this case, the IR diffraction intensity of the grating is decreased when the hand is used to interact with the grating (Supplementary Text, Section 1). The color bar is thus inversed to characterize the generated IR structural colors. When the environmental temperature is the same as the hand temperature, the IR radiation from hand cannot cause the change of IR diffraction intensity of gratings and thus cannot generate different IR structural colors. In this case, we can rub the hand to increase the hand temperature instantly, and thus increase the temperature differences between the hand and the environment (Supplementary Figure 9f).”

We also revised the following statement in Supplementary Text, Section 1 in the revised Supplementary Information to discuss the effect of hand temperature and environmental temperature on the change of IR diffraction intensity of gratings.

Original statement:

“When there is an IR light source, for example, a human hand, the temperature of hand is generally higher than that of the sample. In this case, the grating with higher diffraction efficiency diffracts IR radiation with higher intensity, which increases the detected IR radiance.”

Revised statement:

“When there is an IR light source, for example, a human hand near the gratings, the temperature of hand is generally higher than that of the gratings or the ambient, which means the IR radiation intensity from hand is higher than that from environmental background. In this case, gratings will diffract IR radiation with increased intensity and the grating with higher diffraction efficiency diffracts IR radiation with higher intensity, which increases the detected IR radiance. When the ambient temperature is higher than hand temperature, the IR radiation intensity from hand is lower than that from environmental background. In this case, the IR diffraction intensity of the grating is decreased when the hand is used to interact with the grating, which decreases the detected IR radiance.”

6. In the Introduction, when mentioning pressure-based human-machine interfaces, a classic work could be credited (ACS Nano 2015, 9, 105).

Response: We want to thank the reviewer for the suggestion. This work (Chen, J. et al. Personalized keystroke dynamics for self-powered human-machine interfacing. *ACS Nano* **9**, 105-116 (2015)) is indeed a classic work for pressure-based human-machine interfaces. We have cited it in the Introduction as Reference 16 in the revised manuscript.

7. It is suggested to polish the English writing and reduce the typos. The reference format needs to be carefully revised to fit the journal requirement. Some of the current reference formats are not consistent with each other, for example, some article titles are capitalized, some are not.

Response: Thank the reviewer for the suggestion. We have polished the English writing and revised the typos in the revised manuscript. The reference format has also been revised based on the journal requirement.

Comments from Reviewer #2:

General comment:

This manuscript describes non-contact mode human-machine interface platform in which as a simple demonstration, the motion of a toy vehicle is controlled by hand gestures. The method is based on IR-responsive structural color arising from a periodically patterned metallic grating. More specifically, IR emission from a hand or a finger was interacted with the periodic metal structure, giving rise to the unique structural color in IR (constructive interference between the pattern and IR from the hand). The structural color depends upon not only the characteristics of a pattern such as periodicity and duty cycle but also incident angle of the input IR. By carefully controlling those materials and geometric factors, the authors successfully demonstrate that a finger motion with respect to a periodic pattern is recognized by the different structural color. The structural color in IR regime was converted to visible one via a microprocessor. In addition, the motion-dependent structural color was transferred to the motion switches of a toy vehicle, making the vehicle move around by the programmed hand motion. The authors claim that the system was long-distance responsive from a few centimeters to tens of centimeters, fast responsive, and reliable. In addition, by utilizing IR arising from human body, the motion interaction was done in dark condition, broadening its suitability. The motivation of the work for human-machine interaction is clearly accomplished by the interesting approach based on structural color ascribed to the interaction between a periodic pattern and IR from human body.

Response: We want to thank the reviewer for the encouraging comment. Based on the comments from the reviewer, we have made thorough revisions to address the reviewer's concerns. Following are the point-by-point responses to the detailed comments from the reviewer.

1. Considering that most of works have dealt with visible structural colors for direct visualization of stimuli without additional signal converting technique, the work utilizing IR structural color seems unique. However, at the same time, it is a weakness of the work since the system requires a microprocessor to convert IR structural color to either visible one or sensor signal for triggering the vehicle motion. In this sense, the work should be carefully compared with IR detection and sensing technologies with which human natural IR is readily detected (e.g. IR thermal camera). IR from a finger can be detected by a conventional IR detector as a function of the finger distance and position, and the information of the IR detector can be transferred to either visible display or motion switch of a toy vehicle similar to what the current work demonstrated based on IR structural color. In other words, the various demonstrations shown in the work based on IR structural color could be done with a conventional IR detector. This issue should be clarified.

Response: We want to thank the reviewer for the comment and suggestion. As suggested by the reviewer, we compared IR structural color based-HMI with the conventional IR thermal camera-based HMI. In the HMI composed of IR thermal camera, IR camera is widely used to directly capture the IR images of gestures. The captured gestures are recognized as specific commands for HMI, which can be referred as direct gesture-based HMI. Direct gesture-based HMI can be divided into four stages and each stage has its challenge (Fig. R2-1, left column): (1) Capture of IR images of gestures. The hand is a complex articulated object with more than 20 degrees of freedom. The gestures are thus complex and IR images of gestures contain a large amount of information, which makes real-time HMI challenging. (2) Gesture segmentation from the background. In this stage, the complex background will interfere the segmentation of

gestures and thus affects the recognition performance. (3) Gesture modeling and feature extraction. Different gestures are distinguished from each other by their unique features. The self-occlusion, which means the block or overlap of fingers by other fingers or other parts of the hand, may interfere the extraction of features and thus affects the recognition performance. (4) Gesture recognition as specific command. In this stage, a classifier recognizes gestures as specific commands based on the incoming features. Due to the interference of complex background and self-occlusion in stage 2 and 3, the gesture recognition process is highly dependent on algorithms and the complex algorithms are needed for the recognition process. To overcome these challenges in direct gesture-based HMI, an alternative approach may provide different path for the effective and accurate gesture recognition.

Different from the direct capture and recognition of complex gestures by IR camera, here we used gratings to serve as the interface for the selective interaction with the IR light from fingers, and to convert complex gestures into easily recognizable patterns of IR structural colors (Fig. R2-1, right column). Such selective interaction and conversion process helps minimize the interference of background and the self-occlusion of the fingers. The generated simple and recognizable patterns of IR structural colors also generate much less data to be processed and require less complex algorithms in the analysis process, both of which help simplify the signal recognition process. Following are the more detailed description of comparison that shows the advantages offered by the IR structural color-based HMI presented in this study:

Fig. R2-1. The comparison between HMI composed of the IR camera with direct imaging of gesture and IR structural color-based HMI.

(1) Amount of information captured for processing

In direct gesture-based HMI, real-time recognition is a challenge because gestures are relatively complex and irregular, which makes the captured images of gestures contain large amount of information. The processing of these images is time-consuming. Using the grating array as the human-machine interface, the complex gestures are converted into simple and recognizable color

patterns. The simplification of the recognized patterns can reduce the amount of information of each image, which is beneficial for real-time recognition. Here we use information entropy to quantitatively compare the amount of information between IR images of gestures directly captured by an IR imager and the IR color patterns generated from the hand-grating interaction. The smaller entropy means the less amount of information. In the original manuscript, we demonstrate that we can use individual finger as the IR light source to interact with the grating array and generate different patterns of IR structural colors by vertically moving the finger to the incident angle ranges of different gratings (G_g , G_y , and G_r) (Fig. R2-2a). Here we compared the entropy of two processes (IR structural color-based imaging and direct IR imaging of gestures) by using one finger. As shown in the original manuscript and also in Fig. R2-2a, we can generate three different IR color patterns (p_1 , p_2 , and p_3) by moving the index finger in the IR structural color-based imaging process. To compare with the direct IR imaging of gestures, we also directly captured the IR images of the same gesture at the positions where these three IR color patterns were generated. The directly captured IR images of the gesture at different positions, which are named as g_1 , g_2 , g_3 , are shown in Fig. R2-2b. We then calculated the information entropy of these IR images respectively. As shown in Fig. R2-2c (left), the information entropy of directly captured IR images of gestures (1.42 bits/pixel on average) is much larger than that of IR color patterns (0.16 bits/pixel on average). In the original manuscript, we also demonstrate that we can use the whole hand as the IR light source to interact with the grating array and generate different patterns of IR structural colors. Here we further compared the entropy of two processes (IR structural color-based imaging and direct IR imaging of gestures) by using the whole hand as well. The comparison process is the same as the above comparison process, with the generated IR color patterns named as p_4 , p_5 , and p_6 and the corresponding gestures named as g_4 , g_5 , and g_6 respectively. As shown in Fig. R2-2c (right), the average information entropy of IR color patterns generated by the gesture of the whole hand is 0.16 bits/pixel, also much less than the average information entropy of gesture of the whole hand (1.48 bits/pixel). The small information entropy means that IR color patterns contain much less amount of information in the signal processing, which is beneficial for real-time recognition.

Fig. R2-2. The comparison of information entropy of captured IR images. **a** IR color patterns (p1, p2, and p3) generated by vertically moving the finger to the incident angle ranges of different gratings (Gg, Gy, and Gr). **b** Directly captured IR images of corresponding gestures (g1, g2, and g3) that generate the IR color patterns (p1, p2, and p3) in Fig. R2-2a. **c** The calculated entropy of the IR images in Fig. R2-2a and b (left two columns) and the calculated entropy of IR images by using the whole hand as the IR light source (right two columns). p4, p5, and p6 represent the generated IR color patterns by using the whole hand as the IR light source. g4, g5, and g6 represent the directly captured IR images of corresponding gestures that generate the IR color patterns (p4, p5, and p6).

We added Fig. R2-2 as Supplementary Figure 17 in the revised Supplementary Information and added the following statement in the revised Supplementary Information as Supplementary Text, Section 5 to describe the detailed calculation process of information entropy:

“Section 5. Calculation of two-dimensional (2D) information entropy of captured IR images

2D information entropy is defined by the grayscale distribution of an image, which can characterize the amount of data in the image. For each pixel of the image, the grayscale of the pixel and the average grayscale of its neighborhood are first calculated, which are i and j respectively. The grayscale of the pixel (i) and the average grayscale of the neighborhood (j) form a pair. The probability of each pair is then calculated to be p_{ij} . The 2D entropy can be calculated by the formula:

$$E = - \sum_{i=0}^{255} \sum_{j=0}^{255} p_{ij} \log_2 p_{ij} \quad (18)''$$

We also added a section “Comparison between direct gesture-based HMI and IR structural color-based HMI” in the revised Supplementary Information as Supplementary Text, Section 6 to address the reviewers’ concern and added the following statement in this section to show the difference of the amount of information captured:

“IR camera is also widely used to directly capture gestures. The captured gestures are recognized as specific commands for HMI, which can be referred as direct gesture-based HMI. Compared with direct gesture-based HMI, the conversion of complex gestures into simplified IR color patterns can decrease the amount of information in the captured image and thus is beneficial for real-time recognition. The amount of information of an image can be characterized by information entropy (Supplementary Text, Section 5). The smaller entropy means the less amount of information. In Fig. 3 we demonstrate that we can use individual finger as the IR light source to interact with the grating array and generate different patterns of IR structural colors by vertically moving the finger to the incident angle ranges of different gratings (Gr, Gy, and Gg). Here we quantitatively compared the entropy of two processes (IR structural color-based imaging and direct IR imaging of gestures) by using one finger. As shown in Supplementary Figure 17a, we can generate three different IR color patterns (p1, p2, and p3) by moving the index finger in the IR structural color-based imaging process. To compare with the direct IR imaging of gestures, we also directly captured the IR images of the same gesture at the positions where these three IR color patterns were generated. The directly captured IR images of the gesture at different positions, which are named as g1, g2, g3, are shown in Supplementary Figure 17b. We calculated the information entropy of these IR images respectively. As shown in Supplementary Figure 17c (left), the information entropy of directly captured IR images of gestures (1.42 bits/pixel on average) is much larger than that of IR color patterns (0.16 bits/pixel on average). In Supplementary Figure 7, we also demonstrate that we can use the whole hand as the IR light source to interact with the grating array and generate different patterns of IR structural colors. Here we compared the entropy of two processes (IR structural color-based imaging and direct IR imaging of gestures) by using the whole hand as well. The comparison process is the same as the above comparison process, with the generated IR color patterns named as p4, p5, and p6 and the corresponding gestures named as g4, g5, and g6 respectively. As shown in Supplementary Figure 17c (right), the average information entropy of IR color patterns generated by the gesture of the whole hand is 0.16 bits/pixel, also much less than the average information entropy of gesture of the whole hand (1.48 bits/pixel). The smaller information entropy means that IR color patterns contain less amount of information, which is beneficial for real-time recognition.”

(2) Interference of background

In direct gesture-based HMI, the IR imager captures the IR image of the whole hand and the background. The complex background and the other parts of the hand will interfere the recognition of the gesture (Wang, M. *et al.* Gesture recognition using a bioinspired learning architecture that integrates visual data with somatosensory data from stretchable sensors. *Nat. Electron.* **3**, 563-570 (2020). Chakraborty, B. K. et al. Review of constraints on vision-based gesture recognition for human–computer interaction. *IET Comput. Vis.* **12**, 3-15 (2018)). The direct gesture-based HMI process thus requires the segmentation of gesture from the background (Fig. R2-1).

In comparison, the IR structural color-based HMI uses the grating array as the interface to selectively interact with fingers, which minimizes the interference of background and other parts of the hand. As discussed in the original manuscript, the relationship between incident angle (θ_{in}) and diffraction angle (θ_d) satisfies the diffraction equation:

$$\sin\theta_m = \sin\theta_d - \frac{m\lambda}{\Lambda} \quad (\text{R2-1})$$

where m is the diffraction order and is a nonzero integer, λ is the wavelength of the light, and Λ is the grating period. -1st diffraction efficiency ($m = -1$) in general is higher than diffraction efficiency of other orders, so we focus on -1st diffraction in this work. With the camera fixed at 10° from the axis perpendicular to the center of the gratings, the detected diffraction angle is $\theta_d = 10^\circ$, which means that only the diffracted light with the angle of 10° can be captured and detected by the camera while diffracted light with other angles cannot be captured by the camera. Based on the detected diffraction angle and the grating period, we can calculate the corresponding incident angle according to Equation R2-1, which is an angle range corresponding to the range of detectable IR light wavelength by the camera (7.5 μm ~ 14 μm). Equation R2-1 shows that not all the IR radiation (from hand and from the background) can be detected by the IR camera. Only the IR radiation with incident angle satisfying Equation R2-1 can interact with gratings and then the diffracted light can be detected by the IR camera, which minimizes the interference of IR radiation from other parts of the hand and the complex background. Besides the theoretical analysis, we further ran an experiment to demonstrate the reduction of the interference from the IR radiation emitted by the background and other parts of the hand due to the selective interaction between hand and gratings (Fig. R2-3). The grating used in this experiment is Gr with the period of 25 μm in Fig. 3e of the original manuscript. For the detected diffraction angle ($\theta_d = 10^\circ$), we can calculate the corresponding incident angle range to be 28° ~ 47°.

When we placed the index finger in the calculated incident angle range, the IR radiation from the finger can be diffracted by the grating with the diffraction angle of $\theta_d = 10^\circ$ (Fig. R2-3a). Apparently, IR radiation from the other part of the hand (the palm) and the background can also interact with the grating. To confirm whether the IR radiation from other parts of the hand can interfere the generation of IR color patterns, we used an Al foil with low emissivity (~0.01) to cover the hand except the index finger, which limits the IR radiation from other parts of the hand (Fig. R2-3b). We compared the IR diffraction intensity of the grating with the hand not covered and covered with the Al foil. As shown in Fig. R2-3e, the cover of the hand with Al foil does not change the IR diffraction intensity of the grating, which means that the IR radiation from the other parts of the hand does not interfere the selective interaction between the index finger and the grating. We also placed a heat source with the temperature of ~80 °C at different positions (Fig. R2-3c and d) and compared the IR diffraction intensity of the grating before and after adding the heat source. As shown in Fig. R2-3e, the extra heat source does not change the IR diffraction intensity of the grating, which means that only the IR radiation with the calculated incident angle can selectively interact with gratings and detected by the IR camera. Other IR radiation does not interfere the generation of IR structural colors. The selective interaction between finger and gratings minimizes the interference of complex background and other parts of the hand on the recognition of gestures and commands.

Fig. R2-3. The reduction of interference from background and other parts of the hand due to the selective finger-grating interaction. **a** Placing the index finger in the calculated incident angle range. **b** Covering the palm with the Al foil. **c** Placing a heat source with the temperature of $\sim 80^\circ\text{C}$ above the finger. **d** Placing a heat source with the temperature of $\sim 80^\circ\text{C}$ on the other side of gratings. **e** The IR diffraction intensity of the gratings under the above four conditions. Error bars represent the standard deviation of the mean.

We added Fig. R2-3 as Supplementary Figure 18 in the revised Supplementary Information and added the following statement in the section of “Comparison between direct gesture-based HMI

and IR structural color-based HMI” in the Supplementary Information to address the reviewer’s concern:

“In direct gesture-based HMI, the IR camera captures the IR image of the whole hand and the background. The complex background and the other parts of the hand will interfere the recognition of the gesture. In comparison, this IR structural color-based HMI uses the grating array as the interface to selectively interact with fingers, which minimizes the interference of background and other parts of the hand. To demonstrate such minimized interference due to the selective interaction between hand and gratings, we used Gr with the period of 25 μm (Fig. 3e) to interact with the hand (Supplementary Figure 18). For the detected diffraction angle ($\theta_d = 10^\circ$), the corresponding incident angle range is calculated to be $28^\circ \sim 47^\circ$. When we placed the index finger in the calculated incident angle range, the IR radiation from the finger can be diffracted by the grating with the diffraction angle of $\theta_d = 10^\circ$ (Supplementary Figure 18a). Apparently, IR radiation from the other parts of the hand (the palm) and the background can also interact with the grating. To confirm whether the IR radiation from other parts of the hand can interfere the generation of IR diffraction patterns, we used an Al foil with low emissivity (~ 0.01) to cover the hand except the index finger, which limits the IR radiation from other parts of the hand (Supplementary Figure 18b). We compared the IR diffraction intensity of the grating with the hand not covered and covered with the Al foil. As shown in Supplementary Figure 18e, the cover of the hand with Al foil does not change the IR diffraction intensity of the grating, which means that the IR radiation from the other parts of the hand does not interfere the selective interaction between the index finger and the grating. We also placed a heat source with the temperature of $\sim 80^\circ\text{C}$ at different positions (Supplementary Figure 18c and 18d) and compared the IR diffraction intensity of the grating before and after adding the heat source. As shown in Supplementary Figure 18e, the extra heat source does not change the IR diffraction intensity of the grating, which means that only the IR radiation with the calculated incident angle can selectively interact with gratings and detected by the IR camera. Other IR radiation does not interfere the generation of IR structural colors. The selective interaction between hand and gratings minimizes the interference of complex background and other parts of the hand on the recognition of gestures and commands.”

We also added the following statement in the revised manuscript (Page 11, Paragraph 1, Line 2) for the theoretical analysis of the selective interaction between finger and gratings:

“In the interaction between finger and the grating, the other parts of the hand and the background also emit IR radiation and interact with the grating. Due to the fix of the detected diffraction angle, not all the IR radiation (from hand and from the background) can be detected by the IR camera after the interaction with gratings. Only the IR radiation with incident angle satisfying diffraction equation (Equation 1) can interact with gratings and then the diffracted light is detected by the IR camera. The selective interaction between finger and grating minimizes the interference of IR radiation from other parts of the hand and the complex background on the generation of IR structural colors.”

(3) Interference of self-occlusion

Direct gesture-based HMI generally involves gestures composed of multiple fingers. In this case, self-occlusion is hard to avoid, both in the direct imaging using visible imagers or IR imagers, because the hand is a complex articulated object with more than 20 degrees of freedom (Ge, L. et al. Real-time 3D hand pose estimation with 3D convolutional neural networks. *IEEE Trans. Pattern Anal. Mach. Intell.* 41, 956-970 (2019). Chakraborty, B. K. et al. Review of constraints

on vision-based gesture recognition for human–computer interaction. *IET Comput. Vis.* **12**, 3-15 (2018)). The self-occlusion of gestures will interfere the extraction of features from gestures and thus affect the recognition performance of HMI. In the IR structural color-based process presented in this work, the highly selective interaction between individual finger and the corresponding grating minimizes the impact of self-occlusion, since grating only interacts with the corresponding finger, while other fingers or other parts of the hand will not interfere such interaction, as demonstrated in Fig. R2-3. We further ran an experiment to use two fingers in the two processes (direct IR imaging of gestures and the IR structural color-based imaging) to compare the effect of self-occlusion on the generated IR images. This experiment involves three gestures (g7, g8, g9 in Fig. R2-4a-c) for the direct imaging process. For comparison, in the IR structural color-based process, we used two gratings (Gr and Gg in Fig. 3e in the original manuscript) to interact with these three same gestures. As shown in Fig. R2-4a, we first used the extending middle finger and index finger as the gesture (g7). We then bent the middle finger to 90° to generate a new gesture (g8, Fig. R2-4b). We further bent the middle finger to the palm and generated another gesture (g9, Fig. R2-4c). For the first process that involves the direct capturing of the IR images of the three gestures, we captured the three gestures using IR camera from two different angles (front view and side view). From the captured IR images of gestures (Fig. R2-4d-i) we can find that self-occlusion is hard to avoid and interferes the extraction of features from gestures. For example, in the IR image of g8 captured from the front view (Fig. R2-4e), the bending part of the middle finger is overlapped in this angle (white rectangle), which makes the feature of bending the middle finger unrecognizable in the captured IR image and makes the differentiation of IR images of g8 (Fig. R2-4e) and g9 (Fig. R2-4f) a little challenging. When the IR camera captured gestures from the side view, the bending of the middle finger is recognizable (Fig. R2-4h). In this case, however, it is hard to differentiate the captured IR image of g7 (Fig. R2-4g) from that of g9 (Fig. R2-4i) due to the block of the middle finger by the index finger (blue rectangle). The self-occlusion is thus hard to avoid in the direct gesture-based HMI, which interferes the extraction of features from gestures and the recognition performance of gestures. In comparison, for the IR structural color-based process, we captured the IR color patterns using the same three gestures. We first placed the gesture g7 in the incident angle range of Gr to generate the IR diffraction pattern of “red” on Gr (p7, Fig. R2-4j). We then bent the middle finger to 90° to generate a gesture g8. The bending part of the middle finger covered the incident angle range of Gg and generated the IR diffraction pattern of “green” on Gg (p8, Fig. R2-4k). Meanwhile, the covered incident angle range of Gr decreased (Fig. R2-4b) and thus the “red” on Gr turned to “yellow” (Fig. R2-4k). With the further bending of the middle finger to the palm (Fig. R2-4c), the middle finger cannot cover the incident angle range of Gg and thus “green” turned off in this case (p9, Fig. R2-4l). The selective interaction between fingers and gratings converted features of gestures into patterns of IR structural colors with minimized interference of self-occlusion. As shown in Fig. R2-4j-l, the three IR color patterns can be clearly differentiated based on the color of Gr and Gg. The minimized interference of self-occlusion can enhance the recognition performance of HMI.

Fig. R2-4. The characterization of the interference of self-occlusion on the recognition of features. **a-c** Schematic illustration of three gestures (g7-g9). **d-f** Directly captured IR images of gestures from the front view. **g-i** Directly captured IR images of gestures from the side view. The white rectangle and the blue rectangle show the self-occlusion of gestures, which interferes the extraction of features. **j-l** Generated IR color patterns (p7-p9) by corresponding gestures.

We added Fig. R2-4 as Supplementary Figure 19 in the revised Supplementary Information and added the following statement in the section of “Comparison between direct gesture-based HMI and IR structural color-based HMI” in the Supplementary Information to address the reviewer’s concern:

“Direct gesture-based HMI generally involves gestures composed of multiple fingers. In this case, self-occlusion is hard to avoid because the hand is a complex articulated object with more than 20 degrees of freedom. The self-occlusion of gestures will interfere the extraction of features from gestures and thus affect the recognition performance of HMI. In the IR structural color-based process, the highly selective interaction between individual finger and the corresponding grating minimizes the impact of self-occlusion, since grating only interacts with the corresponding finger, while other fingers or other parts of the hand will not interfere such interaction. We further ran an experiment to use two fingers in the two processes (direct IR imaging of gestures and the IR structural color-based imaging) to compare the effect of self-occlusion on the generated IR images. This experiment involves three gestures (g7, g8, g9 in Supplementary Figure 19a-c) for the direct imaging process. For comparison, in the IR structural color-based process, we used two gratings (Gr and Gg in Fig. 3e) to interact with these three same gestures. As shown in Supplementary Figure 19a, we first used the extending middle finger and index finger as the gesture (g7). We then bent the middle finger to 90° to generate a new gesture (g8, Supplementary Figure 19b). We further bent the middle finger to the palm and generated another gesture (g9, Supplementary Figure 19c). For the first process that involves the direct capturing of the IR images of the three gestures, we captured the three gestures using IR camera from two different angles (front view and side view). From the directly captured IR images of gestures (Supplementary Figure 19d-i) we can find that self-occlusion is hard to avoid

and interferes the extraction of gesture features. For example, in the IR image of g8 captured from the front view (Supplementary Figure 19e), the bending part of the middle finger is overlapped in this angle (white rectangle), which makes the feature of bending the middle finger unrecognizable in the captured IR image and makes the differentiation of IR images of g8 (Supplementary Figure 19e) and g9 (Supplementary Figure 19f) a little challenging. When the IR camera captured gestures from the side view, the bending of the middle finger is recognizable (Supplementary Figure 19h). In this case, however, it is hard to differentiate the captured IR image of g7 (Supplementary Figure 19g) from that of g9 (Supplementary Figure 19i) due to the block of the middle finger by the index finger (blue rectangle). The self-occlusion is thus hard to avoid in the direct gesture-based HMI, which interferes the extraction of features from gestures and the recognition performance of gestures. In comparison, for the IR structural color-based process, we captured the IR color patterns using the same three gestures. We first placed the gesture g7 in the incident angle range of Gr to generate the IR diffraction pattern of “red” on Gr (p7, Supplementary Figure 19j). We then bent the middle finger to 90° to generate gesture g8. The bending part of the middle finger covered the incident angle range of Gg and generated the IR diffraction pattern of “green” on Gg (p8, Supplementary Figure 19k). Meanwhile, the covered incident angle range of Gr decreased (Supplementary Figure 19b) and thus the “red” on Gr turned to “yellow” (Supplementary Figure 19k). With the further bending of the middle finger to the palm (Supplementary Figure 19c), the middle finger cannot cover the incident angle range of Gg and thus “green” turned off in this case (p9, Supplementary Figure 19l). The selective interaction between fingers and gratings converted features of gestures into patterns of IR structural colors with minimized interference of self-occlusion. As shown in Supplementary Figure 19j-l, the three IR color patterns can be clearly differentiated based on the color of Gr and Gg. The minimized interference of self-occlusion can greatly enhance the recognition performance of HMI.”

(4) Simplified algorithm

Due to the complexity of gesture and interference of background and self-occlusion, direct gesture-based HMI includes four stages (Fig. R2-1). Except the capture of gestures by IR camera, other three stages, including gesture segmentation from the background, gesture modeling and feature extraction, and gesture recognition as specific commands are processed by algorithms. The recognition performance of gestures is highly dependent on algorithms. Much effort in the direct gesture-based HMI thus focuses on the development of algorithms.

In comparison, our work uses a grating array to interact with hand. The feature of each gesture is directly converted into easily recognizable patterns of IR structural colors. The use of grating array thus simplifies the algorithm to directly recognize features of images, with no need of gesture segmentation from the background and the hand modeling and feature extraction. As discussed in original Supplementary Information, Section 3, the algorithm just needs to extract the color of each grating and then the corresponding command will be recognized. The simplified algorithm is beneficial for real-time HMI without the need of relatively complex hardware and software for the required algorithms used in the direct gesture-based HMI.

We added the following statement in the section of “Comparison between direct gesture-based HMI and IR structural color-based HMI” in the Supplementary Information to address the reviewer’s concern:

“Direct gesture-based HMI usually includes four stages. Except the capture of gestures by IR camera, other three stages, including gesture segmentation from the background, gesture

modeling and feature extraction, and gesture recognition as specific commands are all processed by algorithms. The recognition performance is thus highly dependent on the algorithms. In comparison, the use of grating array to selectively interact with gestures minimizes the interference of complex background and self-occlusion. The feature of each gesture is directly converted into easily recognizable patterns of IR structural colors. This IR structural color-based HMI thus simplifies the algorithm to recognize features of images, with no need of gesture segmentation from the background and the hand modeling and feature extraction. The simplified algorithm is beneficial for real-time HMI without complex hardware and software.”

We also added the following statement in the revised manuscript (Page 20, Paragraph 2, Line 13) to summarize the comparison result between direct gesture-based HMI and IR structural color-based HMI:

“Comparison between direct-gesture based HMI and IR structural color-based HMI. IR camera is widely used to directly capture the IR images of gestures. The captured gestures are recognized as specific commands for HMI, which can be referred as direct gesture-based HMI. Different from the direct capture and recognition of complex gestures by IR camera, in this work we use gratings to serve as the interface for the selective interaction with the IR light from fingers, and to convert complex gestures into easily recognizable patterns of IR structural colors. We further compare gesture-based HMI and IR structural color-based HMI from four aspects: (1) Amount of information captured for processing. The hand is a complex articulated object with more than 20 degrees of freedom. The gestures are thus complex and IR images of gestures contain a large amount of information, which makes real-time HMI challenging. In contrast, the IR structural color patterns contain much less amount of information, which is beneficial for real-time recognition (Supplementary Text, Section 5 and Supplementary Figure 17). (2) Interference of background. In direct gesture-based HMI, the complex background will interfere the segmentation of gestures and thus affect the recognition performance. In comparison, this IR structural color-based HMI uses the grating array as the interface to selectively interact with fingers, which minimizes the interference of background and other parts of the hand (Supplementary Figure 18). (3) Interference of self-occlusion. In direct gesture-based HMI, different gestures are distinguished from each other by their unique features. The self-occlusion, which means the block or overlap of fingers by other fingers or other parts of the hand, is hard to avoid and interferes the extraction of features. In IR structural color-based HMI, the highly selective interaction between individual finger and the corresponding grating minimizes the impact of self-occlusion, since grating only interacts with the corresponding finger at specific position, while other fingers or other parts of the hand will not interfere such interaction (Supplementary Figure 19). (4) Algorithm. Due to the interference of complex background and self-occlusion, the gesture recognition process is highly dependent on algorithms and complex algorithms are needed for the recognition process. In comparison, IR structural color-based HMI uses a grating array to interact with hand. The feature of each gesture is directly converted into easily recognizable patterns of IR structural colors. The use of grating array thus simplifies the algorithm to directly recognize features of images, with no need of gesture segmentation from the background and the hand modeling and feature extraction. In summary, the selective interaction between hand and gratings in IR structural color-based HMI helps minimize the interference of background and the self-occlusion of the hand. The generated simple and recognizable patterns of IR structural colors also generate much less data to be processed and require less complex algorithms in the analysis process, both of which help simplify the signal recognition process. The detailed comparison result is shown in Supplementary Text, Section 6.”

2. A possible problem of the current approach is that carefully designed pattern structure should be prepared for large area for facile detection of a finger motion. The fabrication of photolithographic patterns seems cost ineffective. In addition, the rigid metallic pattern is hardly altered.

Response: Thank the reviewer for the comment. In the initial demonstration, we used the photolithography to generate the grating patterns. For the large-scale application, other low-cost scalable fabrication processes, for example, soft lithography, may be used. In soft lithography, the initial master pattern can be generated using photolithography or laser microfabrication process. The master pattern is used in the soft lithography process to generate polydimethylsiloxane (PDMS) stamp. The PDMS stamp can be used repeatedly to transfer microstructures onto other substrates, which can lower the cost for large-scale applications. We ran an experiment to generate the PDMS stamp by using a master pattern of the grating array on Si substrate (Fig. R2-5).

Fig. R2-5. Schematic illustration of using soft lithography to fabricate grating arrays.

The fabricated PDMS stamp was then imprinted onto the prepolymer (1-methoxy-2-propyl acetate-based polymer) film coated on Si substrate, which can realize the fast and cost-effective fabrication of grating arrays on Si substrate. Fig. R2-6 shows the fabricated gratings with different structural parameters by soft lithography.

Fig. R2-6. Fabrication of gratings on Si substrate by soft lithography.

To address the reviewer's concern of "In addition, the rigid metallic pattern is hardly altered", we also generated flexible grating arrays. The PDMS stamp generated in the soft lithography process can be used directly as a flexible grating by depositing metal film onto the surface. The period of the grating can be altered by stretching the grating along the direction of period. We

fabricated a PDMS grating with the period of $45\ \mu\text{m}$ and duty cycle of 50%. Based on the period of the grating and the diffraction equation, we can calculate the incident angle range to be $20^\circ\sim 29^\circ$. Placing the index finger in the incident angle range and moving the finger, different IR structural colors can be generated (Fig. R2-7, top). When stretching the sample along the direction of periods, the period of the grating will increase. We stretched the grating by 20%, which means the period of the grating will increase to $54\ \mu\text{m}$. When the finger was placed in the corresponding incident angle range ($18^\circ\sim 26^\circ$), the finger can interact with the grating to generate IR structural colors as well (Fig. R2-7, bottom). When the finger was placed between 27° and 29° , which falls into the original angle range, there was no signal generated.

Fig. R2-7. Fabrication of gratings on flexible PDMS substrate. Top: The generated IR structural colors by placing the index finger in the incident angle range ($20^\circ\sim 29^\circ$) of grating with the period of $45\ \mu\text{m}$. Bottom: The generated IR structural colors after the stretch of the sample by 20%. In this case, the period of the grating increased to $54\ \mu\text{m}$. The finger was placed in the corresponding incident angle range ($18^\circ\sim 26^\circ$) to generate IR structural colors.

We added Fig. R2-5, Fig. R2-6, and Fig. R2-7 as Supplementary Figure 13, Supplementary Figure 14, and Supplementary Figure 15 respectively in the revised Supplementary Information and also the following statement in the revised manuscript (Page 16, Paragraph 1, Line 7) to address the reviewer’s concern:

“The grating array can also be fabricated by soft lithography, in which a flexible (polydimethylsiloxane, PDMS) stamp is first generated and then used to transfer microstructures onto other substrates. The stamp can be used for multiple times and the fabrication is cost-effective for large-scale applications. In this work we also demonstrated the use of soft lithography for the fabrication of grating arrays (Methods and Supplementary Figure 13). As shown in Supplementary Figure 14, gratings with different structural parameters on Si substrate were fabricated by soft lithography. The PDMS stamp can also be used directly as a flexible grating. We fabricated a PDMS grating with the period of $45\ \mu\text{m}$ and duty cycle of 50%. The incident angle range of the grating is $20^\circ\sim 29^\circ$. Placing the index finger in the incident angle range and moving the finger resulted in different IR structural colors (Supplementary Figure 15, top). We stretched the grating along the direction of period by 20%, which means the period of the grating would increase to $54\ \mu\text{m}$ and the corresponding incident angle range would change to $18^\circ\sim 26^\circ$. When the finger was placed in the incident angle range of $18^\circ\sim 26^\circ$, it can interact with the grating to generate IR structural colors as well (Supplementary Figure 15, bottom).”

We also added the following statement in the section of “Methods” in the revised manuscript (Page 24, Paragraph 2, Line 12) to show the fabrication process using soft lithography.

“Fabrication of grating arrays by soft lithography

The fabrication process is shown in Supplementary Figure 13. PDMS precursor was prepared by mixing the silicon elastomer base (Sylgard 184A, Dow Corning Corporation) and the curing agent (Sylgard 184B, Dow Corning Corporation) with the mass ratio of 10:1. The PDMS precursor was poured on the grating arrays fabricated on Si wafer and then cured at 70 °C for 3 h. After curing, the PDMS sample was peeled off from the Si wafer and the inverse structure of the gratings was imprinted on the surface of PDMS. The PDMS sample was then used to fabricate grating arrays on Si substrate. Prepolymer solution (1-methoxy-2-propyl acetate-based, 1 mL) was dropped onto a piece of Si wafer and was spin-coated at the speed of 500 r/min for 1 min. The PDMS stamp was then imprinted onto the prepolymer film to replicate the grating structures on the film. After curing for 4 min at 100 °C, the PDMS stamp was peeled off and the grating structures were generated on the Si substrate. The PDMS stamp itself can be used as a flexible grating as well. Gold (Au) film with thickness of 100 nm was deposited on the PDMS stamp and the polymer gratings on Si substrate by thermal evaporation apparatus to enhance the diffraction efficiency of gratings.”

3. Another issue is the interference of an IR from a finger with various IR sources from environment with different IR wavelengths (heat sources).

Response: We want to thank the reviewer for the comment. This issue is related to the interference from the various IR sources from the environmental background and may also from the other parts of the hand. As we discussed in the response to Question 1, this IR structural color-based HMI uses the grating array as the interface to selectively interact with fingers, which minimizes the interference of background and other parts of the hand. As discussed in the original manuscript, the relationship between incident angle (θ_{in}) and diffraction angle (θ_d) satisfies the diffraction equation:

$$\sin\theta_{in} = \sin\theta_d - \frac{m\lambda}{\Lambda} \quad (\text{R2-1})$$

where m is the diffraction order and is a nonzero integer, λ is the wavelength of the light, and Λ is the grating period. -1st diffraction efficiency ($m = -1$) in general is higher than diffraction efficiency of other orders, so we focus on -1st diffraction in this work. With the camera fixed at 10° from the axis perpendicular to the center of the gratings, the detected diffraction angle is $\theta_d = 10^\circ$, which means that only the diffracted light with the angle of 10° can be captured and detected by the camera while diffracted light with other angles cannot be captured by the camera. Based on the detected diffraction angle and the grating period, we can calculate the corresponding incident angle according to Equation R2-1, which is an angle range corresponding to the range of detectable IR light wavelength by the camera (7.5 μm ~ 14 μm). Equation R2-1 shows that not all the IR radiation (from hand and from the background) can be detected by the IR camera. Only the IR radiation with incident angle satisfying Equation R2-1 can interact with gratings and then the diffracted light can be detected by the IR camera, which minimizes the interference of IR radiation from other parts of the hand and the complex background. Besides the theoretical analysis, we further ran an experiment to demonstrate the reduction of the interference from the IR radiation emitted by the background and other parts of the hand due to the selective interaction between hand and gratings (Fig. R2-8). The grating used in this experiment is Gr with the period of 25 μm in Fig. 3e of the original manuscript. For the detected

diffraction angle ($\theta_d = 10^\circ$), we can calculate the corresponding incident angle range to be $28^\circ \sim 47^\circ$.

When we placed the index finger in the calculated incident angle range, the IR radiation from the finger can be diffracted by the grating with the diffraction angle of $\theta_d = 10^\circ$ (Fig. R2-8a). Apparently, IR radiation from the other part of the hand (the palm) and the background can also interact with the grating. To confirm whether the IR radiation from other parts of the hand can interfere the generation of IR color patterns, we used an Al foil with low emissivity (~ 0.01) to cover the hand except the index finger, which limits the IR radiation from other parts of the hand (Fig. R2-8b). We compared the IR diffraction intensity of the grating with the hand not covered and covered with the Al foil. As shown in Fig. R2-8e, the cover of the hand with Al foil does not change the IR diffraction intensity of the grating, which means that the IR radiation from the other parts of the hand does not interfere the selective interaction between the index finger and the grating. We also placed a heat source with the temperature of $\sim 80^\circ\text{C}$ at different positions (Fig. R2-8c and d) and compared the IR diffraction intensity of the grating before and after adding the heat source. As shown in Fig. R2-8e, the extra heat source does not change the IR diffraction intensity of the grating, which means that only the IR radiation with the calculated incident angle can selectively interact with gratings and detected by the IR camera. Other IR radiation does not interfere the generation of IR structural colors. The selective interaction between finger and gratings minimizes the interference of complex background and other parts of the hand on the recognition of gestures and commands.

Fig. R2-8. The reduction of interference from background and other parts of the hand due to the selective finger-grating interaction. **a** Placing the index finger in the calculated incident angle range. **b** Covering the palm with the Al foil. **c** Placing a heat source with the temperature of $\sim 80^\circ\text{C}$ above the finger. **d** Placing a heat source with the temperature of $\sim 80^\circ\text{C}$ on the other side of gratings. **e** The IR diffraction intensity of the gratings under the above four conditions. Error bars represent the standard deviation of the mean.

We added Fig. R2-8 as Supplementary Figure 18 in the revised Supplementary Information and added the following statement in the section of “Comparison between direct gesture-based HMI

and IR structural color-based HMI” in the Supplementary Information to address the reviewer’s concern:

“In direct gesture-based HMI, the IR camera captures the IR image of the whole hand and the background. The complex background and the other parts of the hand will interfere the recognition of the gesture. In comparison, this IR structural color-based HMI uses the grating array as the interface to selectively interact with fingers, which minimizes the interference of background and other parts of the hand. To demonstrate such minimized interference due to the selective interaction between hand and gratings, we used Gr with the period of $25\ \mu\text{m}$ (Fig. 3e) to interact with the hand (Supplementary Figure 18). For the detected diffraction angle ($\theta_d = 10^\circ$), the corresponding incident angle range is calculated to be $28^\circ\sim 47^\circ$. When we placed the index finger in the calculated incident angle range, the IR radiation from the finger can be diffracted by the grating with the diffraction angle of $\theta_d = 10^\circ$ (Supplementary Figure 18a). Apparently, IR radiation from the other parts of the hand (the palm) and the background can also interact with the grating. To confirm whether the IR radiation from other parts of the hand can interfere the generation of IR diffraction patterns, we used an Al foil with low emissivity (~ 0.01) to cover the hand except the index finger, which limits the IR radiation from other parts of the hand (Supplementary Figure 18b). We compared the IR diffraction intensity of the grating with the hand not covered and covered with the Al foil. As shown in Supplementary Figure 18e, the cover of the hand with Al foil does not change the IR diffraction intensity of the grating, which means that the IR radiation from the other parts of the hand does not interfere the selective interaction between the index finger and the grating. We also placed a heat source with the temperature of $\sim 80\ \text{K}$ at different positions (Supplementary Figure 18c and 18d) and compared the IR diffraction intensity of the grating before and after adding the heat source. As shown in Supplementary Figure 18e, the extra heat source does not change the IR diffraction intensity of the grating, which means that only the IR radiation with the calculated incident angle can selectively interact with gratings and detected by the IR camera. Other IR radiation does not interfere the generation of IR structural colors. The selective interaction between hand and gratings minimizes the interference of complex background and other parts of the hand on the recognition of gestures and commands.”

We also added the following statement in the revised manuscript (Page 11, Paragraph 1, Line 2) for the theoretical analysis of the selective interaction between finger and gratings:

“In the interaction between finger and the grating, the other parts of the hand and the background also emit IR radiation and interact with the grating. Due to the fix of the detected diffraction angle, not all the IR radiation (from hand and from the background) can be detected by the IR camera after the interaction with gratings. Only the IR radiation with incident angle satisfying diffraction equation (Equation 1) can interact with gratings and then the diffracted light is detected by the IR camera. The selective interaction between finger and grating minimizes the interference of IR radiation from other parts of the hand and the complex background on the generation of IR structural colors.”

4. The working principle is based on IR intensity variation depending upon incident angle of IR source from a finger, which seems continuously varied with the incident angle (Figure 3f). First of all, theoretical analysis of this behavior should be provided to confirm the experimental results.

Response: We want to thank the reviewer for the suggestion. The relationship between incident angle (θ_{in}) and diffraction angle (θ_d) satisfies the diffraction equation:

$$\sin\theta_{in} = \sin\theta_d - \frac{m\lambda}{\Lambda} \quad (\text{R2-1})$$

where m is the diffraction order and is a nonzero integer, λ is the wavelength of the light, and Λ is the grating period. The -1st diffraction efficiency ($m = -1$) in general is higher than diffraction efficiency of other orders, so we focus on -1st diffraction in this work. With the camera fixed at 10° from the axis perpendicular to the center of the gratings, the detected diffraction angle is fixed as $\theta_d = 10^\circ$. With the fixed diffraction angle of $\theta_d = 10^\circ$ and grating period (Λ), the incident angle (θ_{in}) is relative to the light wavelength (λ) according to Equation R2-1. The detected wavelength range is $7.5 \mu\text{m} \sim 14 \mu\text{m}$. The calculated incident angle is thus a range rather than a specific value. Each incident angle corresponds to one specific light wavelength. Due to the continuous variation of IR radiation intensity with light wavelength ($7.5 \mu\text{m} \sim 14 \mu\text{m}$) and the correspondence between incident angle and wavelength, IR diffraction intensity/efficiency is continuously varied with the incident angle. Fig. 3f in the original manuscript is the calculated relationship between IR diffraction efficiency (ζ) and incident angle by FDTD. Besides the above theoretical analysis and the calculation result, we further ran an experiment to measure the change of IR diffraction efficiency with incident angle. The experimental setup is shown in Fig. R2-9. We used a heating rod with the temperature of 80°C as an IR light source to measure the relationship between IR diffraction efficiency and incident angle/light wavelength. The experimentally measured result is shown in Fig. R2-10 (top). Fig. R2-10 (bottom) is the calculated result (Fig. 3f in the original manuscript). As shown in Fig. R2-10, the experiment result also shows the continuous change of IR diffraction efficiency with incident angle, similar to the calculated results. The experimentally measured variation trend of IR diffraction efficiency for the three gratings (Gr, Gy, and Gg) is also consistent with the calculated results.

To address the reviewer's concern, we added Fig. R2-9 as Supplementary Figure 6 in the revised Supplementary Information and added Fig. R2-10 as the new Fig. 3f in the revised manuscript.

Fig. R2-9. Experimental setup for the measurement of -1st diffraction efficiency.

Fig. R2-10. The change of diffraction efficiency with the incident angle. Top: The experimentally measured diffraction efficiency (ξ). Bottom: The calculated diffraction efficiency (ξ).

We also revised the following statements in the revised manuscript to address the reviewer’s concern:

Original statement (Page 8):

“For the fixed diffraction angle, the corresponding incident angle can be calculated by Equation 1. Since the detected wavelength range by the IR detector is $7.5\ \mu\text{m}\sim 14\ \mu\text{m}$, the corresponding incident angle is a range rather than a specific value.”

Revised statement (Page 8, Paragraph 1, Line 11):

“For the fixed diffraction angle and specific period, the corresponding incident angle can be calculated by Equation 1, which is related to the light wavelength (λ). Since the detected wavelength range by the IR detector is $7.5\ \mu\text{m}\sim 14\ \mu\text{m}$, the corresponding incident angle is a range rather than a specific value. Each angle corresponds to one specific light wavelength.”

Original statement (Page 14):

“We then used FDTD to calculate the distribution of diffraction efficiency for the three gratings. As shown in Fig. 3f, R1 has the highest diffraction efficiency, followed by R2, and R3 has the lowest diffraction efficiency.”

Revised statement (Page 14, Paragraph 2, Line 11):

“We first experimentally measured the distribution of diffraction efficiency for the three gratings (Supplementary Figure 6 and Supplementary Text, Section 3) and then used FDTD to calculate the diffraction efficiencies for the three gratings. As shown in Fig. 3f, both experimental measurement and theoretical calculation show that diffraction efficiency is continuously varied

with the incident angle. R1 has the highest diffraction efficiency, followed by R2, and R3 has the lowest diffraction efficiency.”

We also added the following statement as Supplementary Text, Section 3 in the revised Supplementary Information to show the detailed measurement process of the diffraction efficiency:

“Section 3. Experimental measurement of the relationship between -1st diffraction efficiency and light wavelength/incident angle.

The diffraction efficiency (ξ) is defined as the ratio of IR diffraction intensity (I_d) to the IR radiation intensity (I_r):

$$\xi = \frac{I_d}{I_r} \quad (4)$$

In this work, when there is no hand or other IR light sources in the surroundings, the IR radiation from environmental background will interact with gratings and be diffracted. When hand or other IR light sources with temperature different from the environmental temperature is placed at specific position, the IR light source will replace the background and interact with gratings. Since the IR radiation intensity of IR light source is different from that of background, the IR diffraction intensity changes. The diffraction efficiency (ξ) can thus be calculated by:

$$\xi = \frac{\Delta I_d}{\Delta I_r} \quad (5)$$

where ΔI_d is the change of IR diffraction intensity with the IR light source placed at specific position. ΔI_r is the change of IR radiation intensity with the IR light source placed at specific position, compared with the IR radiation intensity of background.

According to diffraction equation (Equation 2), for the fixed diffraction angle ($\theta_d = 10^\circ$) and specific grating period A , the incident angle (θ_{in}) is related to the light wavelength (λ). For the detected wavelength range $7.5 \mu\text{m}$ (λ_1)~ $14 \mu\text{m}$ (λ_2), each wavelength corresponds to one specific incident angle.

To measure the relationship between -1st diffraction efficiency (ξ) and light wavelength (λ)/incident angle (θ_{in}), we used a heating rod with the diameter of $d = 4 \text{ mm}$ and the temperature of $T_h = 80 \text{ }^\circ\text{C}$ as an IR light source. The experimental setup is shown in Supplementary Figure 6. The horizontal distance between the grating and light source is $L = 9 \text{ cm}$. The environmental temperature is $T_e = 24.3 \text{ }^\circ\text{C}$.

When the IR light source is placed in the calculated incident angle range, the IR radiation is diffracted by the grating and then detected by the IR detector, which is characterized as the increase of the apparent temperature of grating from T_e to T_d . The increased IR diffraction intensity (ΔI_d) can thus be calculated based on Planck distribution function (Equation 3):

$$\Delta I_d = \int_{\lambda_1}^{\lambda_2} M_\lambda(T_d) d\lambda - \int_{\lambda_1}^{\lambda_2} M_\lambda(T_e) d\lambda \quad (6)$$

Due to the small size of the IR light source, it can only cover a small range of incident angle, corresponding to a small range of light wavelength ($\lambda, \lambda + \Delta\lambda$). Compared with using the environmental background as the IR light source, the increase of IR radiation intensity (ΔI_r) by using the heating rod as the IR light source ($T_h = 80 \text{ }^\circ\text{C}$) can be calculated:

$$\Delta I_r = \int_{\lambda}^{\lambda + \Delta\lambda} M_\lambda(T_h) d\lambda - \int_{\lambda}^{\lambda + \Delta\lambda} M_\lambda(T_e) d\lambda \quad (7)$$

The detected wavelength range ($\lambda, \lambda + \Delta\lambda$) is related to the angle of the IR light source. First, we can calculate the incident angle range ($\theta_{in}, \theta_{in1}$):

$$\tan\theta_m = \frac{L}{h} \quad (8)$$

$$\tan \theta^{n1} = \frac{L}{h_1} \quad (9)$$

$$h_1 = h - d \quad (10)$$

where h is the height of the light source.

The detected wavelength range ($\lambda, \lambda + \Delta\lambda$) is thus can be calculated by the diffraction equation (Equation 2):

$$\lambda = \Lambda \cdot (\sin \theta_{in} - \sin \theta_d) \quad (11)$$

$$\lambda + \Delta\lambda = \Lambda \cdot (\sin \theta_{in1} - \sin \theta_d) \quad (12)$$

Combining Equation 8-12, we can deduce the relationship between the detected wavelength range ($\lambda, \lambda + \Delta\lambda$) and the height of the light source:

$$\lambda = \Lambda \cdot (\sin \arctan \frac{L}{h} - \sin \theta_d) \quad (13)$$

$$\lambda + \Delta\lambda = \Lambda \cdot (\sin \arctan \frac{L}{h-d} - \sin \theta_d) \quad (14)$$

Combining Equation 13, 14, we can find that:

$$\Delta\lambda = \Lambda \cdot (\sin \arctan \frac{L}{h-d} - \sin \arctan \frac{L}{h}) \quad (15)$$

Since $d \ll L$ and $h, \Delta\lambda$ is thus a small value. We simplify the Equation 7 as:

$$\Delta I_r = \int_{\lambda}^{\lambda + \Delta\lambda} M_{\lambda}(T_h) d\lambda - \int_{\lambda}^{\lambda + \Delta\lambda} M_{\lambda}(T_e) d\lambda = \Delta\lambda \cdot (M_{\lambda}(T_h) - M_{\lambda}(T_e)) \quad (16)$$

$$= \Lambda \cdot (\sin \arctan \frac{L}{h-d} - \sin \arctan \frac{L}{h}) \cdot (M_{\lambda}(T_h) - M_{\lambda}(T_e))$$

The relationship between diffraction efficiency (ξ) and light wavelength (λ)/incident angle (θ_{in}) can thus be calculated by combing Equation 6 and Equation 16:

$$\xi(\lambda) = \frac{\int_{\lambda}^{\lambda_2} M_{\lambda}(T_d) d\lambda - \int_{\lambda}^{\lambda_2} M_{\lambda}(T_e) d\lambda}{\Lambda \cdot (\sin \arctan \frac{L}{h-d} - \sin \arctan \frac{L}{h}) \cdot (M_{\lambda}(T_h) - M_{\lambda}(T_e))} \quad (17)$$

In this experiment, we moved the IR light source in the incident angle ranges of three different gratings ($\Lambda = 25 \mu\text{m}, 40 \mu\text{m},$ and $60 \mu\text{m}$ respectively) and recorded the height (h) and the apparent temperature of the grating (T_d). Combining Equation 17 and the data of h and T_d , we can calculate the relationship between diffraction efficiency (ξ) and light wavelength (λ)/incident angle (θ_{in}). To simplify the calculation process, we calculated the relative diffraction efficiency ξ (arbitrary unit, a.u.) of radiation with different wavelength.”

5. Next, the authors should think about how to develop so called “reliable motion sensing” for a specific motion with such an analogue type of intensity variation.

Response: We want to thank the reviewer for the suggestion. To develop the reliable motion sensing, especially in the dark, the grating array can be integrated into a wrist band and attached on the wrist to fix the relative position between fingers and gratings. As a proof-of-concept demonstration, we fabricated a grating array composed of two same gratings with period of $20 \mu\text{m}$

and duty cycle of 50%. As shown in Fig. R2-11a, two gratings are attached on a wrist band and aligned with the middle finger and the ring finger respectively. The two gratings are invisible in the visible image of the wrist band and we use two squares with different colors to show the relative position of the two gratings. The grating corresponding to blue square is named as Gb and the grating corresponding to the violet square is named as Gv. With the bending of middle finger, the IR radiation from the middle finger will selectively interact with Gb. The covered

incident angle range is adjustable by bending the finger with different degrees. Similarly, the ring finger can selectively interact with Gv by bending the finger. As shown in Fig. R2-11b and Movie R1, different IR color patterns are generated with the bending of different fingers. In the future, IR detectors may be also integrated into the wrist band to fix the relative position among fingers, grating array, and IR detectors, which can further enhance the reliability and portability of the HMI system.

Fig. R2-11. Integration of grating array into a wrist band. **a** Two gratings (blue grating (Gb) and violet grating (Gv)) are integrated into a wrist band, which are aligned with the middle finger and the ring finger respectively. **b** The generated IR color patterns with the bending of middle finger/ring finger.

We added Fig. R2-11 as Fig. 4e and 4f in the revised manuscript and added Movie R1 as Supplementary Movie 3 in the revised Supplementary Information. We also added the following statement in the revised manuscript (Page 19, Paragraph 2, Line 15) to address the reviewer’s concern:

“The grating array can also be integrated into a wrist band and attached on the wrist to fix the relative position between fingers and gratings and thus enhance the reliability and operability of HMI in the dark. As shown in Fig. 4e, two same gratings with period of 20 μm and duty cycle of 50% are attached on a wrist band and aligned with the middle finger and the ring finger respectively. The two gratings are invisible in the visible image of the wrist band so we use two squares with different colors to show the relative position of the gratings (Fig. 4e). The grating corresponding to blue square is named as Gb and the grating corresponding to the violet square is named as Gv. With the bending of middle finger, the IR radiation from the middle finger will selectively interact with Gb. The covered incident angle range is adjustable by bending the finger with different degrees. Similarly, the ring finger can selectively interact with Gv by bending the finger. Different IR color patterns are generated with the bending of different fingers (Fig. 4f and Supplementary Movie 3). In the future, IR detectors may also be integrated into the wrist band to fix the relative position among fingers, grating array, and IR detectors, which can further enhance the reliability and portability of the HMI system.”

6. Broad specification of R1, R2, and R3 seems quite vague.

Response: We want to thank the reviewer for the comment. For the three gratings (Gr, Gy, Gg) with the period of 25 μm , 40 μm , and 60 μm , the corresponding incident angle ranges are 28°~47°, 21°~32°, and 17°~24° respectively. We can find that the incident angle ranges are partially overlapped. To realize the selective interaction between finger and different gratings, R1, R2, and R3 should be only within the incident angle range of one grating without overlapping with each other. R1, R2, and R3 are thus specified to be 32°~47°, 24°~28°, 17°~21° respectively.

We added the following statement in the revised manuscript (Page 14, Paragraph 2, Line 6) to address the reviewer's concern:

“The corresponding incident angle ranges for the three gratings are $28^{\circ}\sim 47^{\circ}$, $21^{\circ}\sim 32^{\circ}$, and $17^{\circ}\sim 24^{\circ}$ respectively. We can find that the incident angle ranges are partially overlapped. To realize the selective interaction between finger and different gratings, R1, R2, and R3 should be only within the incident angle range of one grating without overlapping with each other. R1, R2, and R3 are thus specified to be $32^{\circ}\sim 47^{\circ}$, $24^{\circ}\sim 28^{\circ}$, $17^{\circ}\sim 21^{\circ}$ respectively.”

7. The device reliability, detection speed, and resolution and so on should be more quantitatively analyzed.

Response: We want to thank the reviewer for the suggestion. To analyze the device reliability, as suggested by the reviewer, we further calculated the recognition accuracy of the interaction between hand and the wrist band described in Fig. R2-11 in real time. The accuracy was calculated to be 99.2% by generating 24 patterns of IR structural colors in a sequence and repeating the process by 10 times. We also invited operators with different hand size to wear the wrist band and interact with the grating array. As shown in Fig. R2-12, operators with different hand length can interact with grating arrays to generate IR color patterns. The high recognition accuracy and the suitability for different hand size demonstrate the high reliability of the device. As to the device detection speed, we used response time to characterize the detection speed. The response time is the time from the IR camera starting to capture the pattern of IR structural colors to sending the recognized command to the robot vehicle, which is 20.96 ± 1.20 ms. For the device resolution, it depends on the IR intensity resolution, which is represented by the temperature resolution of the IR detector used in the HMI process. The temperature resolution of the IR detector is ~ 0.1 °C, so the device demonstrated here also has a temperature resolution of ~ 0.1 °C.

Fig. R2-12. Generation of patterns of IR structural colors by operators with different hand length (HL).

We added Fig. R2-12 as Supplementary Figure 16 in the revised Supplementary Information and the following statements in the revised manuscript to address the reviewer's concern:

In Page 17, Paragraph 1, Line 22 we added the following statement:

“The IR intensity resolution of the device is represented by the temperature resolution of the IR detector used in the HMI process, which is ~ 0.1 °C. The response time is 20.96 ± 1.20 ms, which is the time from the IR detector starting to capture the pattern of IR structural colors to sending the recognized command to the robot vehicle.”

In Page 20, Paragraph 1, Line 6, we added the following statement:

“The recognition accuracy of the wrist band is 99.2% and the wrist band is applicable for operators with different hand length (Supplementary Figure 16).”

8. Most critically, the work based on IR structural color should be clearly distinguished from conventional non-contact IR detection system.

Response: We want to thank the reviewer for the comment. As suggested by the reviewer, we compared our system with the conventional non-contact IR detection system. The detailed comparison is described in the response to Question 1. In summary, we used gratings to serve as the interface for the selective interaction with the IR light from fingers, and to convert complex gestures into easily recognizable patterns of IR structural colors. Such selective interaction and conversion process helps minimize the interference of background and the self-occlusion of the fingers. The generated simple and recognizable patterns of IR structural colors also generate much less data to be processed and require less complex algorithms in the analysis process, both of which help simplify the signal recognition process.

With the above responses to the reviewers' concerns and revision to the manuscript, we hope that the manuscript can be reassessed for its suitability for publication in *Nature Communications*. Please let us know if any further revision or information is needed to assist your reassessment.

Thanks again for your great help!

Very truly yours,

Tao Deng
Shanghai Jiao Tong University

REVIEWERS' COMMENTS

Reviewer #1 (Remarks to the Author):

The authors well addressed all my previous concerns that I would like to recommend it to be accepted by Nature Communications. This is definitely a piece of exciting work.

Reviewer #2 (Remarks to the Author):

I have carefully read through the rebuttal letter from the authors including the fruitful experimental results and interpretation to address the concerns I raised to make the manuscript further improved. The main concern regarding the benefit of the authors' technique, compared with the conventional IR image technology seems clarified by the authors. A detailed and reasonable comparison of their approach based on IR structural color with the current IR image one has been made. Other issues including large-area fabrication of the patterned structures, theoretical understanding of the IR structural color-based gesture sensing, and reliability of their technique have been also addressed in a proper way. Acknowledged by the authors' considerable efforts for the revision of the manuscript, I am happy to recommend the publication of the work in NCOMM.